# Emergent Bayesian Behaviour and Optimal Cue Combination in LLMs

## Abstract

Large language models (LLMs) excel at explicit reasoning, but their implicit computational strategies remain underexplored. Decades of psychophysics research show that humans intuitively process and integrate noisy signals using near-optimal Bayesian strategies in perceptual tasks. We ask whether LLMs exhibit similar behaviour and perform optimal multimodal integration without explicit training or instruction. Adopting the psychophysics paradigm, we infer computational principles of LLMs from systematic behavioural studies. We introduce a behavioural benchmark - BayesBench: four magnitude estimation tasks (length, location, distance, and duration) over text and image, inspired by classic psychophysics, and evaluate a diverse set of nine LLMs alongside human judgments for calibration. Through controlled ablations of noise, context, and instruction prompts, we measure performance, behaviour and efficiency in multimodal cue-combination. Beyond accuracy and efficiency metrics, we introduce a Bayesian Consistency Score that detects Bayes-consistent behavioural shifts even when accuracy saturates. Our results show that while capable models often adapt in Bayes-consistent ways, accuracy does not guarantee robustness. Notably, GPT-5 Mini achieves perfect text accuracy but fails to integrate visual cues efficiently. This reveals a critical dissociation between capability and strategy, suggesting accuracy-centric benchmarks may over-index on performance while missing brittle uncertainty handling. These findings reveal emergent principled handling of uncertainty and highlight the correlation between accuracy and Bayesian tendencies. We release our psychophysics benchmark and consistency metric as evaluation tools and to inform future multimodal architecture designs.

## 1 Introduction

The estimation of magnitudes, including quantities like length, duration, or distance, represents one of the most fundamental computations in biological and artificial intelligence. Humans perform these judgments through the Bayesian integration of noisy sensory signals, automatically weighting cues by their reliability (Ernst & Banks, 2002) and incorporating prior expectations to minimise estimation error (Remington et al., 2018; Knill & Pouget, 2004). This computational strategy emerges without explicit instruction across diverse cultures and developmental stages, suggesting it reflects a fundamental solution to information processing under uncertainty.

This universality raises the critical question of whether modern LLMs, trained solely on next-token prediction without explicit perceptual objectives (Radford et al., 2018), spontaneously develop analogous computational strategies. Understanding how LLMs process and integrate uncertain information has immediate implications for building robust multimodal systems that appropriately handle varying input quality (Kendall & Gal, 2017; Ma et al., 2022).

To investigate this, we apply classical psychophysics methodology (Petzschner et al., 2015) to probe these implicit computational strategies in LLMs, treating them as black-box observers and inferring their mechanisms from systematic behavioural analysis. By controlling stimulus uncertainty and measuring characteristic signatures of Bayesian processing, we can determine whether LLMs exhibit human-like optimal perception without explicit training. We found that classic identity mapping tasks, prevalent in psychophysics studies, transfer well to experiments with LLMs and reveal a rich set of patterns. We demonstrate that these controlled tasks serve as necessary 'unit tests'

for multimodal robustness. Unlike naturalistic tasks where noise is unquantifiable, our protocol enables controlled ablations that test for optimal integration strategies, exposing brittleness invisible to standard benchmarks. We present three contributions: 1) We introduce a systematic psychophysics framework for LLMs, a reproducible pipeline for four synthetic magnitude estimation tasks probing length, location, distance, and duration. Our pipeline allows controlled ablations of noise, context, and instruction prompts to track behavioural changes. This framework can serve as infrastructure for future investigations bridging human psychophysics studies 2) We develop a new benchmark: BayesBench based on task performance, cue-combination efficiency, and Bayesian consistency computed with a novel Bayesian Consistency Scores 3) We demonstrate emergent Bayes-consistent behaviour in capable LLMs, while uncovering a critical 'Safety Gap' where highly accurate models fail to adopt robust strategies.

## 2 RELATED WORK

**Human psychophysics.** The quantitative study of perception has revealed systematic relationships between physical stimuli and perceptual judgements, formalised in classical laws like Weber-Fechner's logarithmic scaling and Vierordt's temporal regression effects (Fechner, 1860; Weber, 1834; Gibbon, 1977; Jazayeri & Shadlen, 2010; Roseboom et al., 2019; Fountas & Zakharov, 2023). These phenomena, including scalar variability and sequential biases, emerge from optimal Bayesian inference under uncertainty (Petzschner & Glasauer, 2011). When observers estimate magnitudes, they automatically combine noisy measurements with prior expectations, producing characteristic behavioural patterns. Figure 1 illustrates this regression-to-the-mean effect in both Llama-4 Maverick's responses and human psychophysics data—evidence of shared computational principles despite vastly different substrates, as we will see in later sections.

**LLMs and Bayesian behaviour.** Certain aspects of LLMs are shown to be consistent with Bayesian computation. For example, in-context learning can be interpreted as approximate Bayesian inference (Xie et al., 2021) and, in reasoning, Bayesian teaching is shown to improve performance (Qiu et al., 2025). Similarly, LLMs spontaneously segment sequences using Bayesian surprise in ways that correlate with human event perception (Kumar et al., 2023; Fountas et al., 2025). However, most studies probe explicit reasoning or learned behaviours, where models can leverage acquired statistical rules, rather than perceptual tasks that could reveal computational strategies emerging implicitly from pretraining.

**Multimodal studies.** Progress have been rapid in developing multimodal LLMs, alongside this is the deployment of benchmarks such as MMbench (Liu et al., 2024) and SEED-bench (Li et al., 2024) that test multimodal reasoning. However, most of these benchmarks do not cover controlled manipulations of modality specific noise for study-

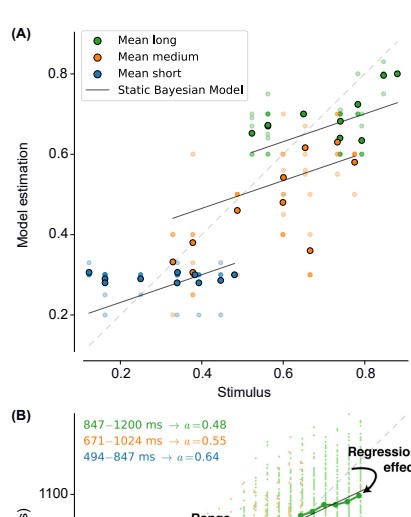

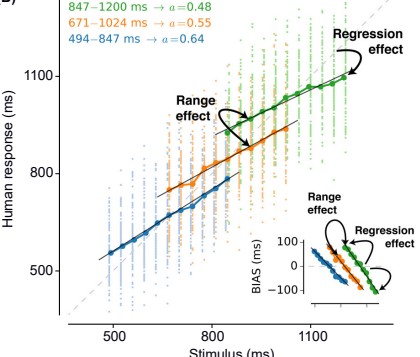

Figure 1: Comparison of LLMs vs human behaviour **A:** Llama-4 Maverick in one of the line length ratio estimation experiments. The fitted lines are based on a static Bayesian observer model. Light dots are individual data points **B:** Response from typical human psychophysics studies (adapted from Thurley, 2016). We see in both that there is a regression to the mean effect, where responses are biased towards the centre of the stimulus range.

ing fusion strategies. Our synthetic datasets allow fine-grained cue-combination analysis and studies how LLMs combine noisy information from multiple modalities. This is still a nascent area of research but crucial for better understanding how we may build more robust and generalisable models that will behave optimally under uncertainty.

## 3 METHODS

### 3.1 ESTIMATION TASKS AND ABLATIONS

We develop four psychophysics-inspired magnitude–estimation tasks illustrated in Figure 2:

- **Marker location estimation:** given a line with a red marker (or '0' in text input) estimate the position of a marker on a line as a number between 0 to 1.
- **Line ratio estimation:** given two lines, estimate the ratio of the shorter line to the longer line.
- **Maze distance estimation:** given a non-self-intersecting path, estimate the straight line distance between start and the end of the path.
- **Duration estimation:** given an extract of a conversation transcript, estimate the duration of the dialogue. Transcripts are extracted from the AMI Meeting Corpus Kraaij et al. (2005).

The first three tasks are multimodal, with text input and image inputs.

We conduct ablations to probe LLMs and analyse changes in behaviours (see Appendix A.3 for ablation details):

- **Steering:** provide additional textual or numerical information in the system prompt. Aimed at studying how LLMs behaviour changes when asked to consider uncertainty in its responses.
- **Noise:** add constant or gradually increasing blur to the image modality. Aimed at studying how LLMs may reweight information in the presence of noise. We chose gaussian noise for its tractability and ubiquity.
- **Context:** change the length of the available history or reversing trial sequence. Aimed at studying how previous context affects behaviour.

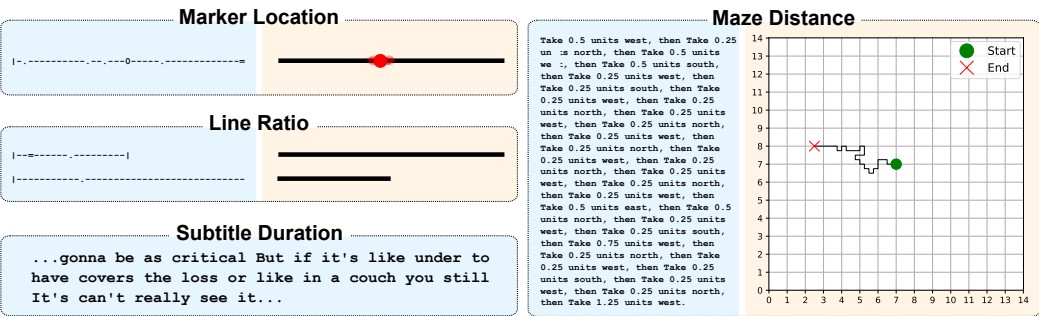

Figure 2: Example of the four magnitude estimation tasks. Cues in a blue background represent information provided as text, while orange represents vision.

### 3.2 BEHAVIOURAL MODELLING

In human psychophysics studies, participants' responses are fitted against a range of behavioural models to infer their internal computational strategies (Petzschner & Glasauer, 2011; Jazayeri & Shadlen, 2010). This is an effective approach when the subject is essentially a black box, and we can only observe their input-output behaviour. In line with this framework, we fit LLMs' responses against a set of behavioural models covering factors of interest. The degree of fit to different models indicates the extent to which LLMs exhibit that behaviour. Note that while we report model evidence against different behavioural models, we do not rely on individual goodness-of-fit in static conditions to demonstrate Bayesian consistent behavior. Instead, these are use as probes for the Bayesian Consistency Score to study how behaviour changes when experimental conditions are manipulated.

In the below, $x_t$ and $y_t$ denote the true input value of the stimulus and the LLM's estimate at trial $t$, respectively. $\mu_t$ and $\sigma_{\text{dec}}$ are the LLM's internal estimate and response noise level, respectively. We used three main types of behaviour models:

**Linear observer.** Linear stimuli-estimation relationship. As our experiments use identity-mapping tasks, many non-probabilistic heuristics (such as regression to any anchor or fixed biases) are captured under this relationship

$$\mu_t = wx_t + b, \quad y_t \sim \mathcal{N}(\mu_t, \sigma_{\text{dec}}^2). \tag{1}$$

**Static Bayesian observer.** LLM's estimation is a weighted average of the input stimulus $x_t$ and a fixed prior belief $\mu_p$:

$$\mu_t = \frac{\tau_x}{\tau_x + \tau_p}x_t + \frac{\tau_p}{\tau_x + \tau_p}\mu_p, \quad y_t \sim \mathcal{N}(\mu_t, \sigma_{\text{dec}}^2), \tag{2}$$

$\tau_x$ and $\tau_p$ denote the measurement and prior precisions respectively. We show in the upper panel of Figure 1 an example where this model best fits the LLM's responses.

**Sequential Bayesian observer (Kalman filter).** LLM's estimation is updated trial-by-trial following a standard Kalman filter:

$$\mu_{t|t-1} = \mu_{t-1|t-1}, \quad P_{t|t-1} = P_{t-1|t-1} + q, \quad y_t \sim \mathcal{N}(\mu_t, \sigma_{\text{dec}}^2), \tag{3}$$

Where the update equations are:

$$K_t = \frac{P_{t|t-1}}{P_{t|t-1} + r}, \quad \mu_{t|t} = \mu_{t|t-1} + K_t(x_t - \mu_{t|t-1}), \quad P_{t|t} = (1 - K_t)P_{t|t-1}. \tag{4}$$

$r$ is the measurement noise variance, $q$ is the process noise variance and $P$ is the variance about its estimate. We show in Figure 3 and 4 an example where this model best fits the LLM's responses. This sequential model is intended to capture within-session inference in a noisy environment. Note that when a model has very high accuracy, response will show little evidence of regression as these are identity mapping tasks. This reinforces the idea that regression effect alone is insufficient and thus the need to introduce alternative probes such as BCS to detect Bayesian behaviours.

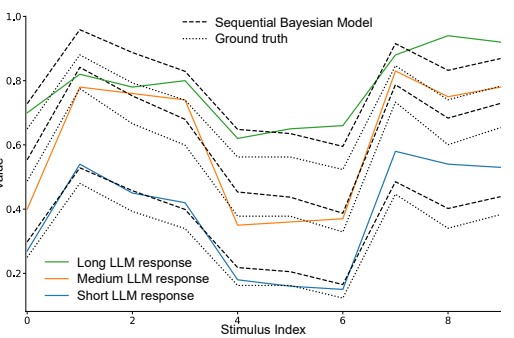
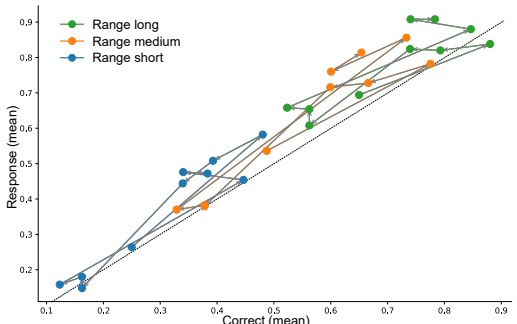

Figure 3: GPT-5 Mini's mean response (to verbal cues) compared to prediction based on a sequential Bayes model (dotted line)

Figure 4: GPT-5 Mini's mean response trajectory (verbal cue). Arrows denote the sequence of its responses.

**Additional model variants** Across all models, we include variants with an additional stage of log-transform on the input and output (this mimics studies that support evidence of a logarithmic perception of magnitude in humans and animals (Nieder & Miller, 2003; Nover et al., 2005).)

For non-linear models, we fitted variants where a final stage of gain or affine transformation is applied. This is to account for potential mis-calibration in output mapping (this is not needed for linear models as it is captured in the bias and gradients). Further details can be found in Appendix A.13.

### 3.3 CUE COMBINATION MODELLING

For our multimodal tasks, we study how LLMs combine text and image cues by modelling their multimodal responses against their unimodal responses. The main models are in Table 1. $y_{\text{comb}}$,

| Equal weighting | Linear regression | Bayes-optimal fusion |
|---|---|---|
| $y_{\text{comb}} = \frac{1}{2}(y_{\text{text}} + y_{\text{image}})$ | $y_{\text{comb}} = \alpha\, y_{\text{text}} + (1 - \alpha)\, y_{\text{image}}$ | $y_{\text{comb}} = w_{\text{text}}\, y_{\text{text}} + (1 - w_{\text{text}})\, y_{\text{image}}$ |
| | | $w_{\text{text}} = \dfrac{1/\sigma_{\text{text}}^2}{1/\sigma_{\text{text}}^2 + 1/\sigma_{\text{image}}^2}$ |

Table 1: Cue-combination baselines. $\alpha$ is fitted in $[0, 1]$. $\sigma_{\text{text}}^2$ and $\sigma_{\text{image}}^2$ are the empirical variances of the LLM's responses in the text-only and image-only conditions respectively.

$y_{\text{text}}$ and $y_{\text{image}}$ denote the LLM's response for multimodal, unimodal text and unimodal image, respectively.

For the *Bayes-optimal fusion* model, we report *Oracle* (calibrated, covariance-based) and *Non-Oracle* (uncalibrated, variance-based) variants. This fusion is the optimal *linear unbiased* combiner (BLUE) under linear-Gaussian assumptions. See Appendix A.7 for details.

### 3.4 MODEL EVIDENCE

Model fit is based on Akaike Information Criterion (AIC). See Appendix A.8 for further details.

### 3.5 KEY METRICS

We quantify model along two dimensions: capability as measured by (i) task accuracy and (ii) cue–combination efficiency, and behaviour strategy as measured by (iii) behavioural consistency. Separation between capability and behaviour strategy allow us demonstrate cases when they dissociate.

**Accuracy (NRMSE).** $\text{NRMSE} = \text{RMSE}_{\text{LLM}}/\text{RMSE}_{\text{baseline}}$, where $\text{RMSE}$ has the standard root-mean-squared-error definition. The baseline is a constant predictor that outputs the mean of the stimulus range (lower is better).

**Efficiency (RRE).** $\text{RRE}(m_{\text{ref}}) = \text{NRMSE}_{\text{ref}}/\text{NRMSE}_{\text{LLM}}$ for any reference combiner $m_{\text{ref}}$ (Sec. 3.3). RRE values $> 1$ ($< 1$) mean the LLM has lower (higher) error than the reference. Note that RRE is used a measure of capability, as it is quantifies the LLM's performance relative to a normative baseline, rather as a proof of behaviour or computation strategy.

**Bayesian Consistency Score (BCS).** To test whether LLM's behaviour shifts in the *Bayes–consistent* direction under controlled ablations, we compare the fitted weights of a static Bayesian observer model (Sec. 3.2). The posterior mean of this model is precision–weighted with $w_{\text{prior}} = \tau_p/(\tau_p + \tau_x)$ (prior precision $\tau_p$, measurement precision $\tau_x$), so increasing $\tau_p$ or decreasing $\tau_x$ raises $w_{\text{prior}}$. Studying changes in behaviour allow us to compare models which may have very different static behaviour.

We use five ablations across three tasks to compute BCS. These ablations are designed to increase $\tau_p$ and/or decrease $\tau_x$: (i) **Steering (verbal)** and (ii) **Steering (unbiased numerical)** provide range–consistent context or prompt the model about measurement noise, effectively strengthening the prior ($\tau_p \uparrow$); (iii) **Noise (constant)** and (iv) **Noise (gradual)** blur image inputs to reduce measurement precision ($\tau_x \downarrow$); (v) **Context (longer context window)** supplies a longer rolling history without altering current measurements ($\tau_p \uparrow$, $\tau_x$ unchanged).

For each ablation $a$, we compare fitted weights to the base experiment, $\Delta w_{\text{prior}} = w_{\text{prior}}^{(\text{ablation})} - w_{\text{prior}}^{(\text{base})}$, and set

$$s_a = \begin{cases} +1 & \text{if } \Delta w_{\text{prior}} \geq 0, \\ -1 & \text{if } \Delta w_{\text{prior}} < 0, \end{cases} \quad \text{with } s_a = 0 \text{ if } w_{\text{prior}}^{(\text{ablation})} > 0.9.$$

We focus on the *sign* of $\Delta w_{\text{prior}}$ since magnitudes depend on model–specific factors, such how accurate or noisy a given model's perception is. For example, a highly perceptually accurate model may only need to adjust $w_{\text{prior}}$ by a smaller amount given an injection of measurement noise. We set $s_a$ to zero when $w_{\text{prior}}^{(\text{ablation})} > 0.9$, because this indicates a prior-dominant regime, where the model is

| Factor | Expression | Parameters |
|--------|-----------|------------|
| NRMSE (A) | $1 - \frac{\text{NRMSE} - \text{NRMSE}_{\min}}{\text{NRMSE}_{\max} - \text{NRMSE}_{\min}}$ | $\text{NRMSE}_{\min,\max} = 0,\ 2$ |
| RRE (E) | $\left[\text{RRE(Bayes Oracle)} + \text{RRE(Bayes Non Oracle)}\right]/2$ | N/A |
| BCS (C) | $(\text{BCS} - \text{BCS}_{\min})/(\text{BCS}_{\max} - \text{BCS}_{\min})$ | $\text{BCS}_{\min,\max} = -15,\ 15$ |

Table 2: BayesBench components. $\text{NRMSE}_{\max}$ is set to 2 (twice the error committed by the constant predictor baseline). $\text{BCS}_{\min,\max}$ are set equal to the range of scores for five ablations across three multimodal experiments.

essentially disregarding the current stimulus and always outputting a constant. This is undesirable because in all five selected ablations the stimulus should remain informative.

The *Bayesian consistency score* sums over ablations: $\text{BCS} = \sum_a s_a$.

### 3.6 COMPOSITE BENCHMARK SCORE (BAYESBENCH).

The overall *BayesBench score* is a function of three metrics: NRMSE factor for task accuracy (A), RRE factor for cue-combination performance against a Bayes-optimal reference (E) and BCS factor for Bayes-consistency in behaviour adaptation (C) (defined in Table 2). The first factor is averaged across all four tasks while the latter two are averaged across the three multimodal tasks. In the (A) factor, $\text{NRMSE}_{\max} = 2$ defines the upper bound of model NRMSE and models that incur larger error receive no credit. This range spans our model range and marks the worst reasonable performance of any model.

The *BayesBench score* is defined as:

$$S_{\text{BayesBench}} = \tfrac{1}{3}\left(A + E + C\right). \tag{5}$$

BayesBench is designed to provide a holistic summary covering both capability and behaviour across tasks.

## 4 EXPERIMENTAL SETUP

Each estimation task is divided into three sessions (short, medium, long), where stimulus values fall into different but overlapping ranges per session. The stimuli are generated from a range-uniform prior. This choice enables a clean separation between prior and likelihood and aligns with decades of human psychophysics studies, where range-uniform priors are standard. The overlap in ranges allows us to study context-dependent effects. Figure 5 provides an overview of the stimulus value distributions across sessions, using the marker location as an example. In each session, at each trial, the LLM is given the context of its prior trials (i.e., both the stimulus probes and the LLM's previous responses, as each API interaction is stateless or "memoryless"). The rolling context simulates how humans form memory of recent interactions, and is the basis of the emergence of Bayesian consistent behaviour. The overall view of our experimental setup is shown in Appendix A.1.

Interactions with LLMs are performed via API. See Appendix A.2 for further details.

We evaluate a diverse set of recent LLMs spanning closed- and open-weight releases (see Appendix A.4 for details). Where possible, we disable extended-thinking or reasoning controls to probe the models' natural, emergent behaviour. This was feasible for all models except GPT-5 mini, which only allows adjusting reasoning depth; we set this to the lowest level.

In addition, we ran a human baseline study for comparison on all our tasks under a small number of ablations. See Appendix A.5 for details. Human results are included in the left panel of Figure 8. This experiment and analysis are used only as a reference point here, as our main focus is on comparing LLMs against each other. Extensive human psychophysics studies, including the magnitude estimation effects examined here, are extensively documented in psychophysics literature, such as in Jazayeri & Shadlen (2010); Petzschner & Glasauer (2011).

We estimate uncertainty in our results using 30 rounds of bootstrapping while preserving the trial structure within each session to maintain contextual integrity. The error bars shown in Figure 8 and 9 represent 68% bootstrap percentile intervals.

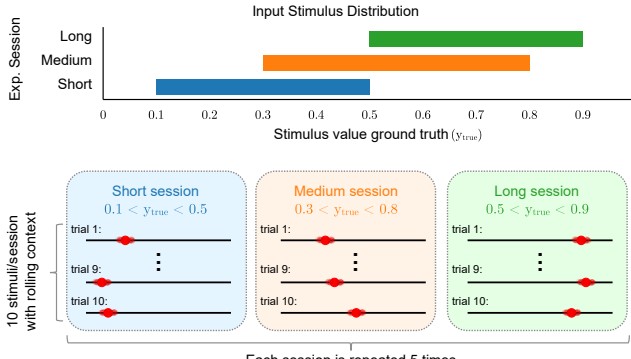

Figure 5: Example distribution of stimulus input for the marker location task

## 5 RESULTS

### 5.1 OVERALL PERFORMANCE AND BEHAVIOURAL FIT

Most models perform better in the *text* than the *image* modality, except on the maze distance estimation task. The text input for the maze distance estimation task is a long, detailed path description—much more complex than the ASCII prompts used in other tasks (Fig. 2). Most models perform worse in the text modality for this task, but GPT-5 Mini is an outlier here: it achieves near-perfect text performance, due to the residual reasoning which we can attenuate but not fully disable and shows the task-dependent nature of behaviour. Across tasks, the factor evidence for Bayesian behaviour is consistently higher in the image modality than in text (Appendix A.11).

Moving from unimodal to multimodal inputs does not uniformly improve performance. However, some models are better able to leverage information from the additional modality: Llama-4 Maverick attains its best performance under multimodal conditions across all tasks, and Claude 3.7 Sonnet and GPT-4o improve on two of the three tasks.

Overall, the strongest models (GPT-5 Mini, Claude 3.7 Sonnet, GPT-4o) reach low error rates, comparable to—or better than—human performance (left panel, Fig. 8).

With the exception of Gemini 2.5 Flash Lite, there is a general trend in the left panel of Figure 8 that more accurate models also show stronger evidence of Bayesian behaviour. Note that the relationship between accuracy and behaviour is empirical and is not by necessity. For example, a perfect predictor that maps all stimulus to accurate estimates is indistinguishable from a linear model and can show no probabilistic tendencies.

Under steering ablation, when a numerical range for prior observations is provided in the context prompt, we find that LLMs behavior is strongly affected. As shown in the top right panel of Figure 6, evidence for sequential behavior decreases strongly and in particular, when a deliberately biased numerical range is given, as shown in the left panel of Figure 6, error increases (the triangle icons are generally at larger NRMSE than the circles). This is consistent with the fact that LLMs gravitate their predictions towards prior information provided in-context.

### 5.2 CUE COMBINATION

From the middle panel of Figure 8, we see that not all models with good NRMSE performance also exhibit efficient cue combination. GPT-5 Mini, despite its strong NRMSE performance, shows poor cue combination efficiency. This is especially pronounced in the maze distance estimation task, where GPT-5 Mini's performance in the text modality is essentially perfect and much better than

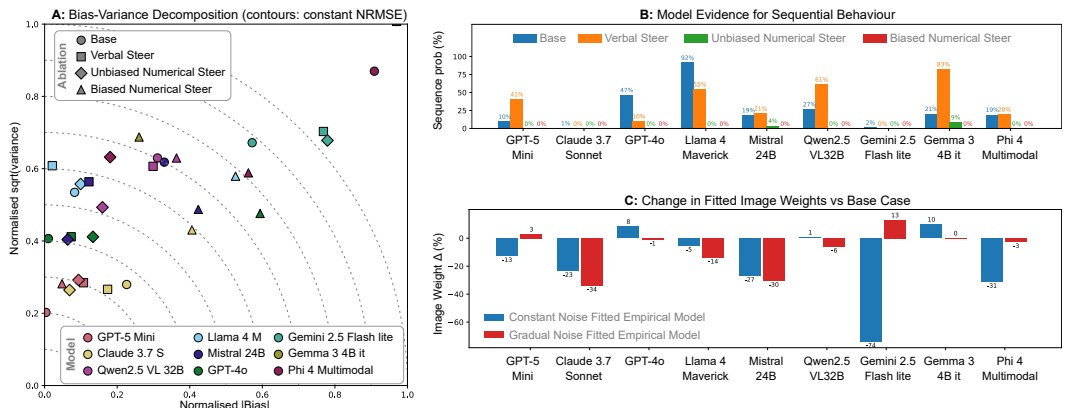

Figure 6: **A:** Bias-variance decomposition under steering ablations. **B:** Impact on sequential model evidence under steering ablation. **C:** Implied image weighting under noise ablation.

its image modality performance. This implies that a Bayes-optimal combination must significantly further downweight its image input. However, it appears unable to downweight its image input to the optimal extent (see Appendix A.12 for further details). On the other hand, in the line length ratio task (see bottom right panel of Figure 6), many LLMs are able to downweigh the image modality in the presence of noise, indicating Bayes-consistent adaptation.

Llama-4 Maverick's multimodal NRMSE performance exceeds that of a Bayesian reliability-weighted unbiased linear combination. The Bayesian reliability-weighted combiner is a normative baseline widely used in biological studies and humans are shown to employ consistent mechanisms Ernst & Banks (2002). Llama-4 Maverick's outperformance indicates a cue-combination efficiency beyond some biological systems including humans. This suggests that the model may be leveraging additional non-linear properties not assumed under our linear baseline. In Figure 7, we fitted Llama-4 Maverick's multimodal responses against its unimodal responses. We found that a non-linear random forest is indeed better able to fit its multimodal responses from unimodal responses than linear variants.

Under linear-Gaussian noise, the Bayesian cue combiner is optimal and thus provides a natural yardstick for future improvement on LLMs that have not yet reached this level of performance.

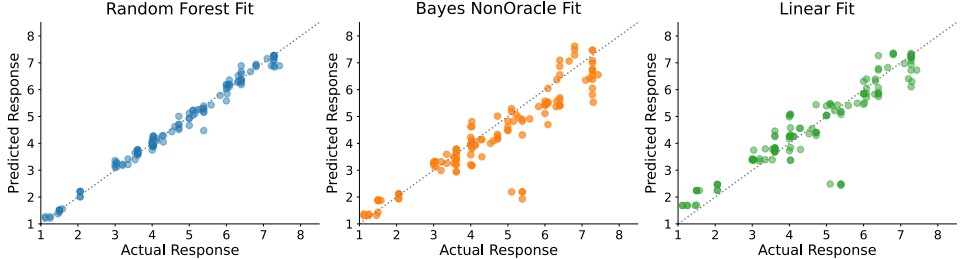

Figure 7: Comparison of cue combination model fits for Llama-4 Maverick. Left panel: random forest fit (blue). Middle panel Bayes-optimal fit (orange). Right panel: linear regression fit (orange).

### 5.3 BAYESIAN CONSISTENCY

From the right panel of Figure 8, we see that generally more accurate models also tend to exhibit more Bayes-consistent behaviour. However, despite Gemma 3 4B and Phi 4 Multimodal's lower accuracy, they achieved a decent BCS value. We show a breakdown of the BCS by task and model in A.9.

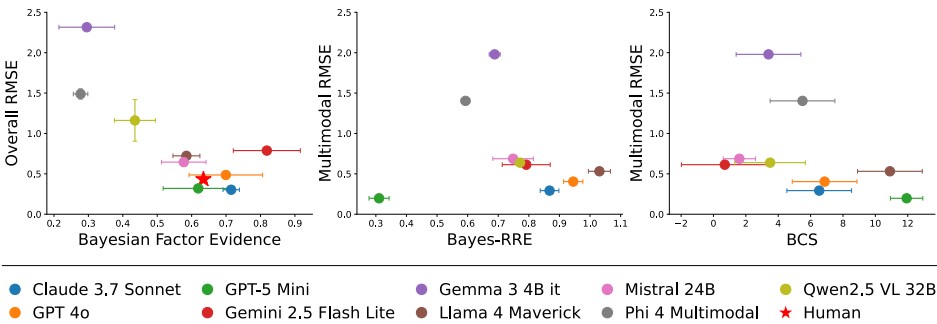

Figure 8: Results summary across models and tasks. Left panel: Bayesian behavioural evidence and relationship against overall NRMSE. Middle panel: cue-combination performance. Shows relationship between multimodal tasks NRMSE and efficiency against Bayes-optimal cue-combination reference models. Right panel: Bayes-consistency score and its relationship against multimodal NRMSE. Each point represents a model, with color indicating model family. Error bars represent 68% bootstrap percentile intervals. Human baseline is shown in the left panel for reference.

## 5.4 BAYESBENCH SUMMARY

Figure 9 shows the computed BayesBench scores across models, in accordance to the definition in Section 3.6. Bayes-RRE generally increases with accuracy (lower NRMSE), with two notable exceptions: GPT-5 Mini underperforms on Bayes-RRE relative to its NRMSE, whereas Llama-4 Maverick exceeds expectations on Bayes-RRE. BCS likewise tends to track accuracy but provides additional separation among the top models. Overall, Llama-4 Maverick attains the highest Bayes-Bench score, driven by strong Bayes-RRE and BCS components.

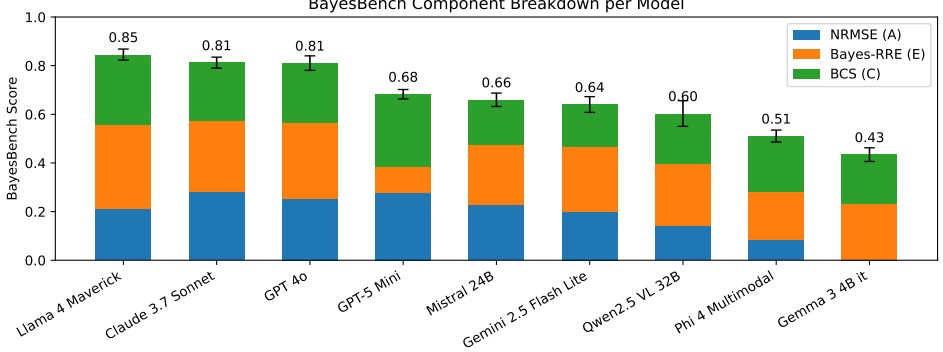

Figure 9: BayesBench overall score, with breakdown into components. Error bars represent 68% bootstrap percentile intervals.

## 6 DISCUSSION

Our study reveals that LLMs exhibit rich and diverse behavioural patterns when probed with psychophysics-inspired magnitude estimation tasks. While the degree of factor evidence for Bayesian behaviour differs by task and modality, more accurate models (e.g., GPT-5 Mini, Claude 3.7 Sonnet, Llama-4 Maverick) tend to display higher Bayesian factor evidence, especially in the image modality (Appendix A.11 for full breakdown). These models tend to adapt their behaviour in Bayes-consistent ways when inputs are subjected to perturbations such as noise, steering, or extended context (right panel of Figure 8) and take into account contextual prior information during estimation even without explicit training or reasoning instructions. This is reminiscent of findings in human psychophysics, where Bayesian models explain a wide range of human perceptual phenom-

ena and processing in the brains (Petzschner et al., 2015; Knill & Pouget, 2004) without explicit training. We emphasise that our analysis at this stage is behavioural and to offer a mechanistic or causal account of how such behaviour are implemented would require further analysis.

We find that high task accuracy does not always imply optimal cue–combination (middle panel of Figure 8). For example, GPT-5 Mini attains very low NRMSE yet does not combine modalities efficiently compared to other models. This shortfall is most apparent when unimodal performance is imbalanced: optimal behaviour would require the model to markedly down-weight the weaker modality, which some LLMs fail to do. This has direct practical implications as it exposes the risk that benchmarks which solely focus on accuracy may favour models with less robustness against noise. Conversely, Llama-4 Maverick surpasses Bayesian reliability-weighted linear fusion, indicating the use of more sophisticated non-linear integration strategies, consistent with the fact that a non-linear random forest fits its cue combination better than linear variants.

Comparing uni- and multimodal performance reveals that, while models such as Llama-4 Maverick, Claude 3.7 Sonnet, and GPT-4o are able to utilise the additional modality of input to achieve lower error when both modalities (text and image) are present for the the majority of multimodal tasks, this is not a universal trend. The variability in gains indicates potential headroom for advancing multimodal LLMs. See Appendix A.11 for model-specific breakdown.

To capture behavioural features beyond static task metrics, we devised the Bayesian Consistency Score (BCS) that captures principled behavioural shifts. This allows us to evaluate model behaviour more holistically, even when accuracy saturates. Measuring behaviour changes under controlled ablations enable us to compare models that may have different base performance and can offer additional insights into implicit computational strategies. While more complex heuristics (e.g., salience-based weighting or rule-switching strategies) may fit static patterns of responses better, it is more revealing to study Bayes-consistent behaviours under a Bayesian observer model when conditions change.

While LLM behaviours are nuanced and context dependent, our results show that LLMs are generally consistent with Bayesian observer models. This raises the question of how Bayesian consistent behaviour can be an emergent property of sufficiently capable models trained on large-scale data, similar to questions tackled in human studies (Barlow et al., 1961; Wei & Stocker, 2015). Future architectures or training regimes that better encode uncertainty and support principled cue combination may improve LLMs' robustness in noisy, real-world settings. Furthermore, benchmarks such as our custom BayesBench can complement standard accuracy-based evaluations, offering diagnostic insights into implicit computational strategies.

**Limitations.** As the test range of our tasks is bounded, effects that only emerge with longer sequences may not be detected. Our ablation studies are necessarily limited in scope; other perturbations may illustrate different aspects of behaviour. In addition, all interactions relied on API access, which may be affected by API non-determinism or silent vendor updates.

## 7 CONCLUSION AND FUTURE DIRECTIONS

We present BayesBench, a psychophysics-inspired benchmark that probes LLMs' ability to estimate magnitudes, integrate noisy multimodal cues, and exhibit Bayes-consistent behaviour. Our findings show that capable LLMs not only achieve low error rates but also adapt in Bayesian consistent manners, revealing emergent cognitive-like strategies. Strong multimodal models can also combine cues efficiently, although this is not guaranteed by high accuracy alone. Our results suggest that Bayesian-consistent behaviour may emerge naturally in sufficiently capable models.

Our work bridges human psychophysics and AI research, by providing both an extensible template and a set of diagnostic metrics. While our tasks are synthetic, they highlight possible directions for studying implicit computation in LLMs. The BayesBench framework could be extended to more naturalistic settings, providing a scaffold for future benchmarks that probe principled cue combination and uncertainty handling in scenarios closer to real-world use. Future work should explore representational underpinnings from a mechanistic perspective, and assess how Bayesian tendencies scale with model size, training data and training objectives.

**Reproducibility Statement** We will release *BayesBench* for public use, including the synthetic data generator, prompts, ablation configurations, behavioural/cue–combination model code and evaluation scripts. The behavioural models are fully specified in Section 3.2; cue–combination models are fully specified in Section 3.3; factor-evidence computation in Section 3.4 and Appendix A.8; the metrics and composite score is specified in Section 3.5 and Appendix A.10;

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

## A  APPENDIX

### A.1  EXPERIMENTAL DESIGN

The basic setup of our experiments follows: 1) dataset and prompt generation, 2) session structure covering the order of stimulus presentation, 3) interation with LLM via API and 4) analyses. This modular design (Figure 10) allows for systematic exploration of different factors influencing model performance.

### A.2  INTERACTION WITH LLMS

#### A.2.1  PROGRAMMATIC API

Our interaction with LLMs is through programmatic API calls with temperature fixed at 0.7. As our aim is to probe the natural, emergent behaviour of highly performant LLMs, we instruct models not to use reasoning or chain-of-thought, returning only the final numeric answer with minimal output text. For GPT-5 Mini, reasoning cannot be disabled, so it is set to the lowest reasoning level available. This provides an additional point of comparison, as reasoning-enabled models may behave differently in textual tasks. This is something we see in our experiments involving GPT-5 Mini.

We emphasise the use of API-based LLMs, some of which are closed source, to ensure our pipeline is lightweight and easily extended to new models.

To test modality dependence, we run tasks in text-only, image-only, and text+image conditions. In text-only mode, line-ratio and marker-location tasks are represented using ASCII, while the maze task is described concisely in text. In image-only mode, models receive only the visual stimulus. In multimodal mode, both text and image inputs are given. This allows us to evaluate efficiency in unimodal vs multimodal contexts.

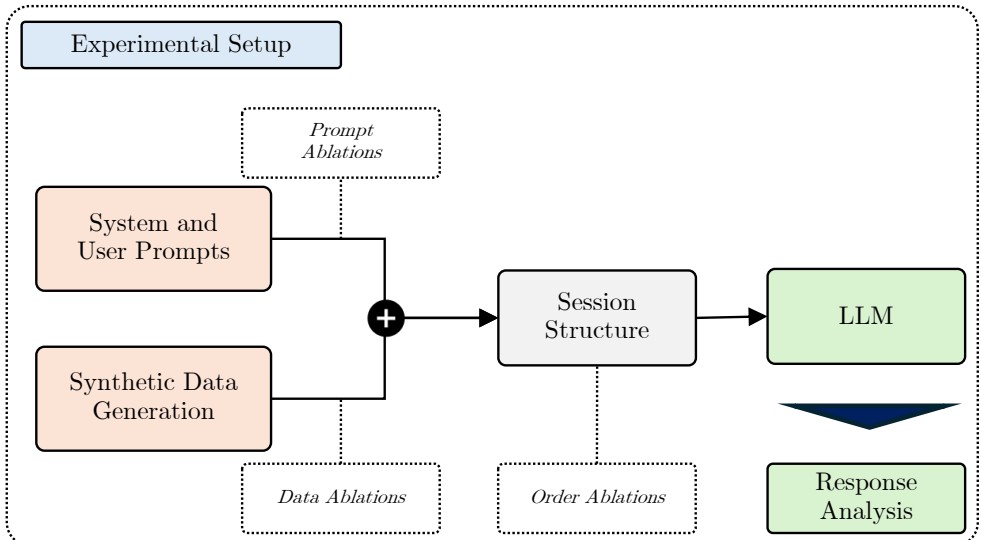

Figure 10: Experimental setup overview

### A.2.2 PROMPT DESIGN

For all tasks, prompts are structured in two parts: a **system prompt** and a **user query**.

- **System prompt**: defines the role of the model (e.g., "You are a line-length ratio estimator."). It specifies the expected output format and instructs the model to *not output reasoning*, but to return only the final numeric estimate (with minimal text if necessary).
- **User query**: provides the stimulus in the chosen modality. In textual mode this is ASCII input (for line ratio and marker tasks) or a concise text description (for maze and subtitle tasks). In image mode only the stimulus image is shown. In multimodal mode both text and image are provided.

Responses that are ill-formed are discarded when we record experimental data.

A typical prompt for the line-length ratio task (textual mode) is:

```
System prompt:  "You are a line-length ratio
estimator.  Estimate the ratio of the shorter line
to the longer line as a decimal number between 0 and
1.  Do not explain or reason.  Only output the final
answer."
User input:

|-=-=------                                          |
|-------------------------.------ -------|
```

This design keeps task specification clear and minimises variation in output. For GPT-5 Mini, where reasoning cannot be disabled, we used the lowest reasoning setting. This provides an additional point of comparison, since reasoning-enabled models may behave differently in textual tasks.

For **steering-related ablations**, modifications are made at the system prompt stage. Models may be told that observations are noisy, or given numerical information about the range of past observations. Further details of these manipulations are described in Section A.3.1.

### A.3 ABLATION BACKGROUND

Ablation conditions are grouped into three categories: steering-related, noise-related, and context-related. Each modifies the base setup in a controlled way to test specific hypotheses.

### A.3.1 STEERING-RELATED ABLATIONS

**Verbal cues**

- Modified the system prompt to explicitly tell the model that observations are noisy and that it should act in a Bayesian way.
- Example system prompt:

    You are a line-length ratio estimator. The given data is noisy and may contain artifacts. You should behave like a Bayesian observer and take into account prior and likelihood in your predictions.

**Numerical cues**

- Modified the system prompt to provide the numeric range of the past ten observations, encouraging the model to use this information as a prior.
- Example system prompt:

    You are a line-length ratio estimator. The given data is noisy and may contain artifacts. For 10 previous observations, the values were observed to lie in the range of 0.1 to 0.3.

### A.3.2 NOISE-RELATED ABLATIONS

**Constant noise:**  Applied a Gaussian blur to image inputs only, to test whether models adapt estimation behaviour when vision is degraded.

**Noise sequence:**  Introduced gradually increasing Gaussian blur across trials to test whether models downweight visual information as noise grows. Figure 12 shows example input images.

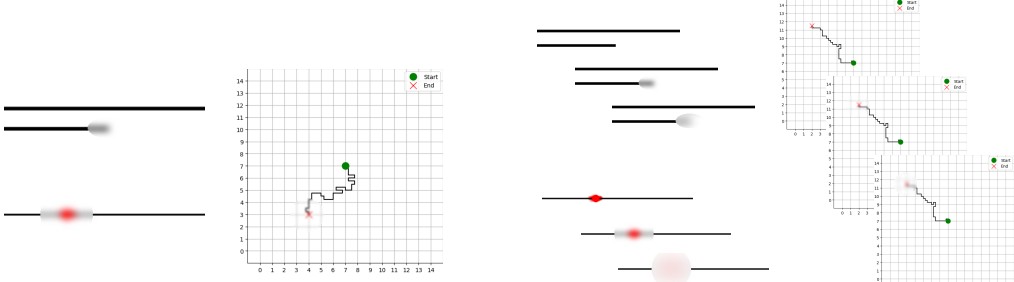

Figure 11: Constant Gaussian noise ablation      Figure 12: Sequential Gaussian noise ablation

### A.3.3 CONTEXT-RELATED ABLATIONS

**Shorter Context:**  Reduced the context window to 3 prior trials, limiting how much past information the model can use.

**Longer context**  Increased the context window to 20 prior trials, maximising available history for the model.

**Stimulus order reversal**  Reversed the order of stimuli to test whether model estimations show strong sequence dependence.

### A.4 LLM MODELS STUDIED

Table 3 summarises the key characteristics of the LLMs studied. We chose a diverse set of recent models spanning closed- and open-weight releases, with a range of sizes and architectures. Where

h

Table 3: Comparison of selected LLMs (parameters shown only when vendor/model card publicly discloses them).

| Model | Developer | Params | Reasoning controls |
|---|---|---|---|
| Claude 3.7 Sonnet | Anthropic | Undisclosed | Optional "extended thinking" |
| GPT-5 Mini | OpenAI | Undisclosed | Adjustable depth. |
| GPT-4o | OpenAI | Undisclosed | N.A. |
| Llama-4 Maverick | Meta | 400B total / 17B active | N.A. |
| Qwen 2.5 VL 32B | Alibaba | 32B | N.A. |
| Mistral 24B | Mistral | 24B | N.A. |
| Gemini 2.5 Flash Lite | Google DeepMind | Undisclosed | N.A. |
| Phi 4 Multimodal | Microsoft | Undisclosed | N.A. |
| Gemma 3 4B | Google DeepMind | 4B | N.A. |

**Notes:** We avoid speculative parameter estimates. Public sources: Claude 3.7 Sonnet announcement (Anthropic); GPT-5 Mini (OpenAI docs); Llama-4 Maverick active/total params (Meta); Qwen 2.5-VL 32B model card; Mistral 24B (Mistral docs); Gemini 2.5 Flash-Lite (Google); Phi-4 Multimodal (Microsoft HF card); Gemma 3 model card.

possible, we disabled extended-thinking or reasoning controls to probe the models' natural, emergent behaviour. This was feasible for all models except GPT-5 Mini, which only allows adjusting reasoning depth; we set this to the lowest level.

### A.5 HUMAN FEEDBACK COLLECTION

We collected data from human subjects on our main tasks to establish a calibration benchmark. The questions are hosted on a web platform, and users can complete them with their phone or computer.

Only two ablations were used for human feedback collection: constant noise and longer context.

Figure 13 shows two screenshots of the web platform.

### A.6 ETHICS STATEMENT

Our human baseline study involved a minimal-risk web-based questionnaire where participants provided perceptual judgments on magnitude estimation tasks. Participation was voluntary, participants provided informed consent via the web platform, could withdraw at any time, and all data was anonymized. No personal, sensitive, or identifiable information was collected.

### A.7 BAYES CUE COMBINATION MODELS

Under Bayesian assumptions, the optimal linear combination of two noisy modality estimates is obtained by weighting them according to their relative reliabilities (inverse variances. See also Ernst & Banks (2002)). We consider two versions. Non-oracle and oracle models. They differ in whether the cue combination is modelled with or without access to ground truth.

- **Non-oracle:** The model combines the two modality estimates ($\mu^{(1)}$ and $\mu^{(2)}$) by inverse-variance weighting,

$$\mu = \frac{\tau_1}{\tau_1 + \tau_2}\mu^{(1)} + \frac{\tau_2}{\tau_1 + \tau_2}\mu^{(2)},$$

where $\tau_i = 1/\sigma_i^2$ are the precisions of estimates from the corresponding modality. Crucially, the model does not assume access to ground truth. It only uses the variance of each modality estimate to compute the above weighting.

- **Oracle:** In this case, we first calibrate the modality-specific estimates ($\mu^{(1)}$ and $\mu^{(2)}$) by fitting gain and offset parameters to the ground truth for each modality. After calibration, the estimates ($\mu'^{(1)}$ and $\mu'^{(2)}$) are combined using the generalised least squares solution

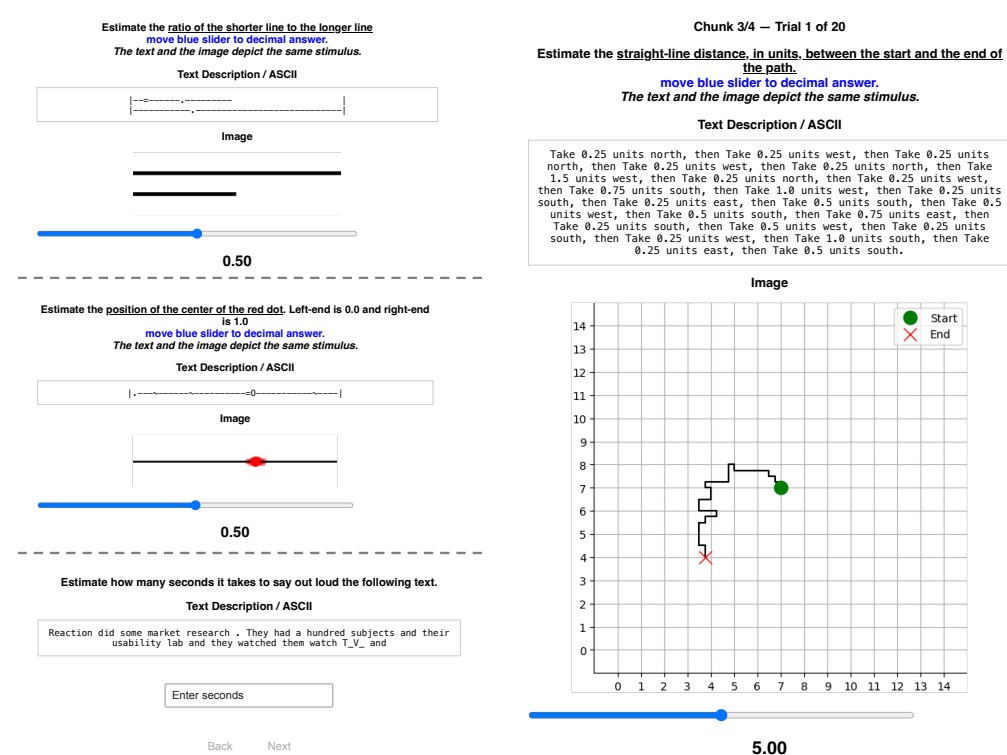

Figure 13: Human feedback collection website screenshot

based on the residual loss covariance ($\Sigma$):

$$\mu = \frac{\mathbf{1}^\top \Sigma^{-1} \boldsymbol{\mu}'}{\mathbf{1}^\top \Sigma^{-1} \mathbf{1}},$$

where $\boldsymbol{\mu}' = \begin{bmatrix} \mu'^{(1)} \\ \mu'^{(2)} \end{bmatrix}$ and $\Sigma$ is the $2 \times 2$ covariance matrix of the modality estimates. This accounts for both differing reliabilities and cross-modal correlations, yielding the optimal linear unbiased estimator given access to the true values.

Although these models specify optimal *linear* integration strategies, it is important to note that LLMs may, in principle, outperform these baselines if they achieve more flexible, nonlinear forms of cue integration. Such nonlinear integration is possible given the architecture of modern LLMs.

A.8  FACTOR ANALYSIS DETAILS

We fit many behavioural model variants that differ along interpretable *factors* (e.g., BAYESIAN vs NON-BAYESIAN; WEBER vs NON-WEBER; SEQUENTIAL update). Because these variants partly overlap in purpose, naively summing or averaging likelihoods would (i) reward families that contain more variants, or (ii) dilute good variants by pooling with weak ones. We therefore compare *factors* while treating all other dimensions as *nuisance*.

**Procedure**  Let $f \in \{\text{BAYESIAN}, \text{WEBER}, \text{SEQUENTIAL}\}$ be the factor of interest, and let $\mathcal{N}(f)$ denote the set of nuisance factors for this comparison (chosen to be agnostic to $f$; see example below).

1. **Transform AIC to likelihood.** For each fitted variant $m$, first compute $\Delta\text{AIC}(m)$ (defined as the difference between $m$'s AIC and the minimum AIC among all variants) and then

compute the transformed quantity below:

$$L(m) \; \propto \; \exp\!\big(-\tfrac{1}{2}\,\Delta\mathrm{AIC}(m)\big),$$

2. **Group by nuisance "cells".** Group behavioural models by every combination of values in $\mathcal{N}(f)$. Each group is a cell $c$.

3. **Best-in-cell for each level of $f$.** Within each cell $c$, take the *maximum* likelihood among variants where $f = \mathtt{True}$ and among variants where $f = \mathtt{False}$:

$$L_{\mathrm{True}}^{(c)} = \max_{m \in c,\; f(m)=\mathtt{True}} L(m), \qquad L_{\mathrm{False}}^{(c)} = \max_{m \in c,\; f(m)=\mathtt{False}} L(m).$$

Using the max avoids penalising a family for having many weak sub-variants.

4. **Equal-weight across cells.** For fairness, average *equally* across cells where both levels are present (intersection):

$$\bar{L}_{\mathrm{True}} = \frac{1}{|C|} \sum_{c \in C} L_{\mathrm{True}}^{(c)}, \qquad \bar{L}_{\mathrm{False}} = \frac{1}{|C|} \sum_{c \in C} L_{\mathrm{False}}^{(c)},$$

where $C = \{c :\; L_{\mathrm{True}}^{(c)}, L_{\mathrm{False}}^{(c)} \text{ both defined above in step 3.}\}$.

5. **Compute evidence.** Report the factor-level probability

$$P(f{=}\mathtt{True} \mid \mathrm{data}) \;=\; \frac{\bar{L}_{\mathrm{True}}}{\bar{L}_{\mathrm{True}} + \bar{L}_{\mathrm{False}}},$$

and similarly for $\mathtt{False}$.

In this report when we refer to ***factor evidence***, we are always referring to evidence computed from this procedure.

**Example: BAYESIAN VS NON-BAYESIAN.** For $f = $ BAYESIAN we take $\mathcal{N}(f) = \{\text{WEBER}\}$ only. The SEQUENTIAL and GAIN variants exist exclusively within the Bayesian family; conditioning on them would create empty cells on the non-Bayesian side. Thus, within each WEBER cell we compare the best Bayesian variant (possibly sequential/gain/log) against the best non-Bayesian variant, average equally over cells, and form the head-to-head probability. Below table shows the procedure schematically.

|  | WEBER cell | |
| --- | --- | --- |
|  | False | True |
| Best Bayesian in cell | $L_{\mathrm{True}}^{(c)}$ | $L_{\mathrm{True}}^{(c)}$ |
| Best non-Bayesian in cell | $L_{\mathrm{False}}^{(c)}$ | $L_{\mathrm{False}}^{(c)}$ |

*Average equally across cells, then compute* $P = \bar{L}_{\mathrm{True}}/(\bar{L}_{\mathrm{True}} + \bar{L}_{\mathrm{False}})$.

**Notes on fairness and robustness.** (i) Equal cell weighting prevents families with many variants from accruing more probability mass simply by proliferation. (ii) Using the intersection of cells avoids bias from missing combinations.

## A.9 BCS BREAKDOWN BY EXPERIMENT

We show below the breakdown of BCS score by experiment and task.

## A.10 BCS FITTING DETAILS

We fit the static Bayesian observer model in all cases and with data from modalities according to the below:

- **Noise:** evaluate $w_{\mathrm{prior}}$ from the *image-only* modality, since noise is injected only into the image channel and multimodal fits would confound reweighting of text input.

- **Steering and Context:** evaluate $w_{\mathrm{prior}}$ from the *multimodal* fit, as these manipulations affect both modalities.

| Model | Line Ratio | Marker Location | Maze Distance |
|---|---|---|---|
| Claude 3.7 Sonnet | 1.5 | 5.0 | 0.0 |
| GPT 4o | 0.8 | 4.5 | 1.5 |
| GPT-5 Mini | 5.0 | 5.0 | 1.9 |
| Gemini 2.5 Flash Lite | 1.7 | -1.3 | 0.3 |
| Gemma 3 4B it | 1.1 | 0.8 | 1.5 |
| Llama 4 Maverick | 3.2 | 4.4 | 3.3 |
| Mistral 24B | 1.5 | 1.0 | -0.9 |
| Phi 4 Multimodal | 2.4 | 1.3 | 1.8 |
| Qwen2.5 VL 32B | -2.0 | 5.0 | 0.5 |

Table 4: Model-wise BCS score across experiments

## A.11 MODEL PERFORMANCE AND BAYESIAN FACTOR EVIDENCE

Figures 14, 15, 16 and 17 show the NRMSE performance and Bayesian factor evidence for all models across all tasks and modalities. For the multimodal tasks, in their corresponding figures, metrics by modality is shown over the three rows.

Notice that not all models perform better in multimodal conditions than in unimodal conditions (Llama-4 Maverick is the outlier, it achieves its best NRMSE in multimodal mode on all tasks).

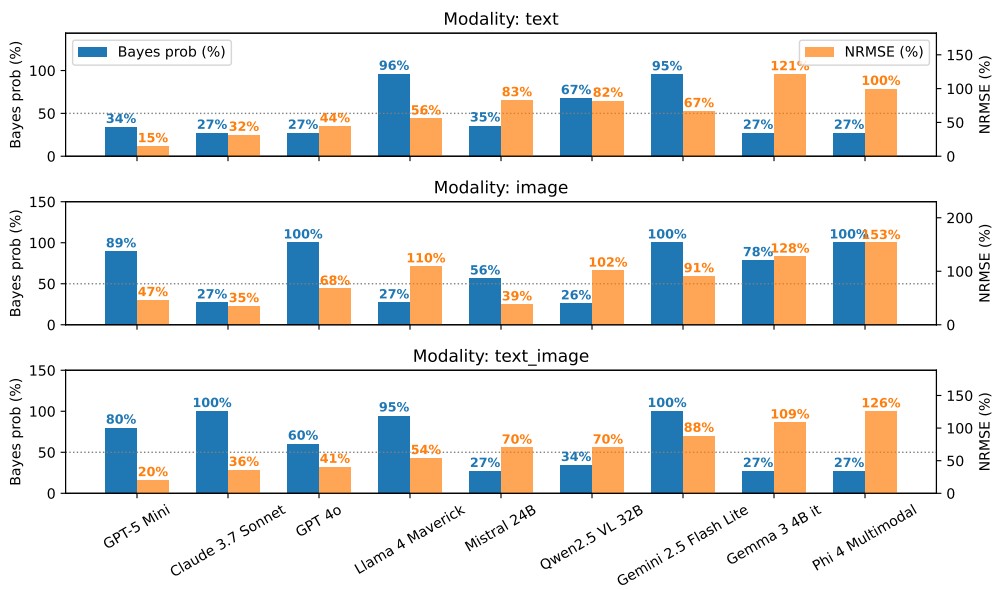

Figure 14: Line length ratio estimation task. NRMSE and Bayes factor evidence for unimodal text, unimodal image and multimodal inputs.

## A.12 GPT-5 MINI CUE COMBINATION MODEL FITS

GPT-5 Mini's cue-combination performance is poor despite its very strong NRMSE performance. Figure 16 shows the NRMSE performance for each model in all three modalities for the maze distance estimation task. We see that GPT-5 Mini's unimodal text performance is nearly perfect (at 0.01 NRMSE), while its unimodal image performance is much worse (at 0.2 NRMSE, despite already being the best across models). Because of this, the Bayes-optimal linear combination would imply a nearly zero weighing on the image input. However, the multimodal performance does not follow this trend, indicating that the model prediction is still affected by the image input.

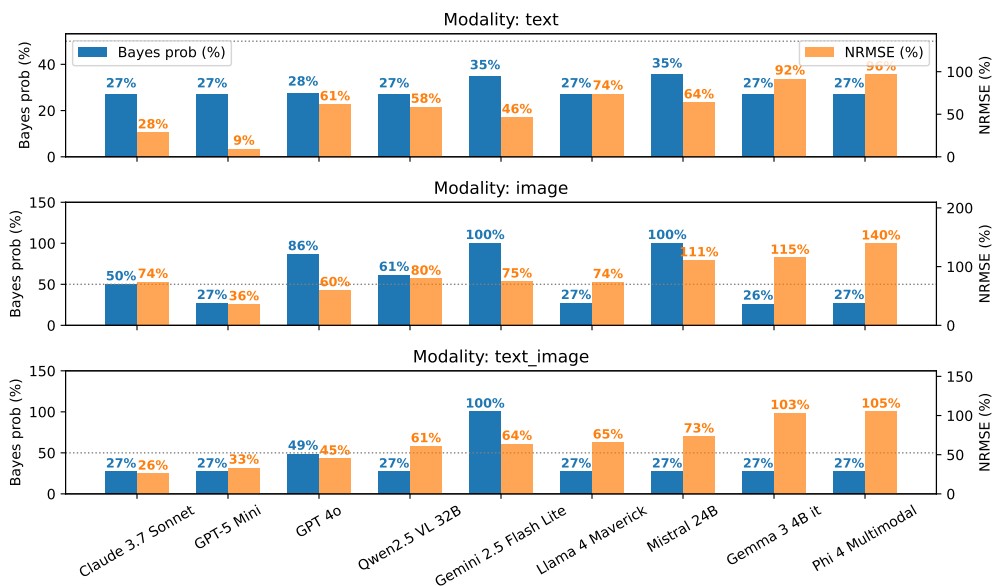

Figure 15: Marker location estimation task. NRMSE and Bayes factor evidence for unimodal text, unimodal image and multimodal inputs.

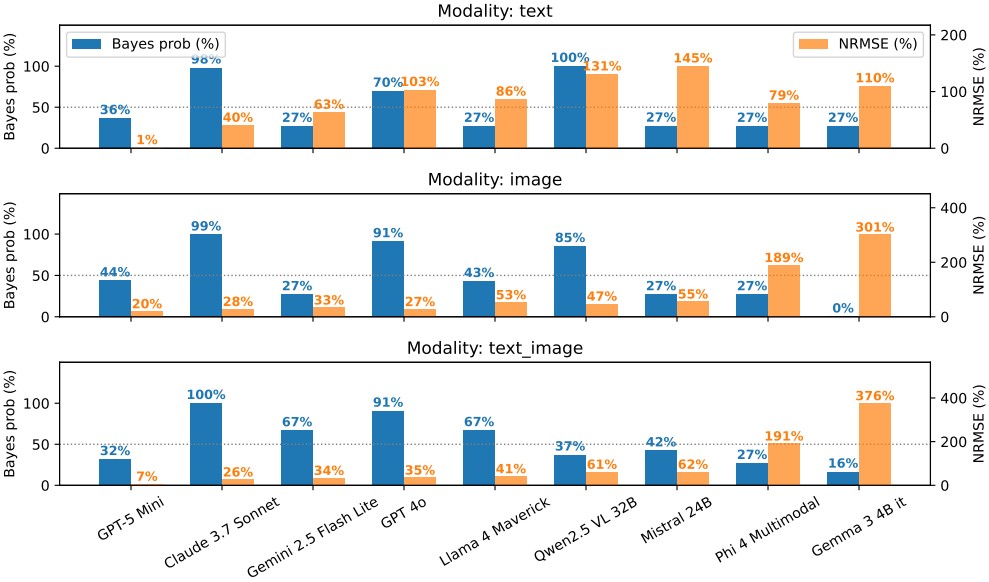

Figure 16: Maze distance estimation task. NRMSE and Bayes factor evidence for unimodal text, unimodal image and multimodal inputs.

## A.13 FURTHER MODEL VARIANTS

Studies such as (Nieder & Miller, 2003; Nover et al., 2005) found in human studies that the human brain encodes many different magnitudes using a logarithmic scale. To test if this phenomena apply in LLM, we explored variants of models where a logarithmic transform is applied to the stimulus values.

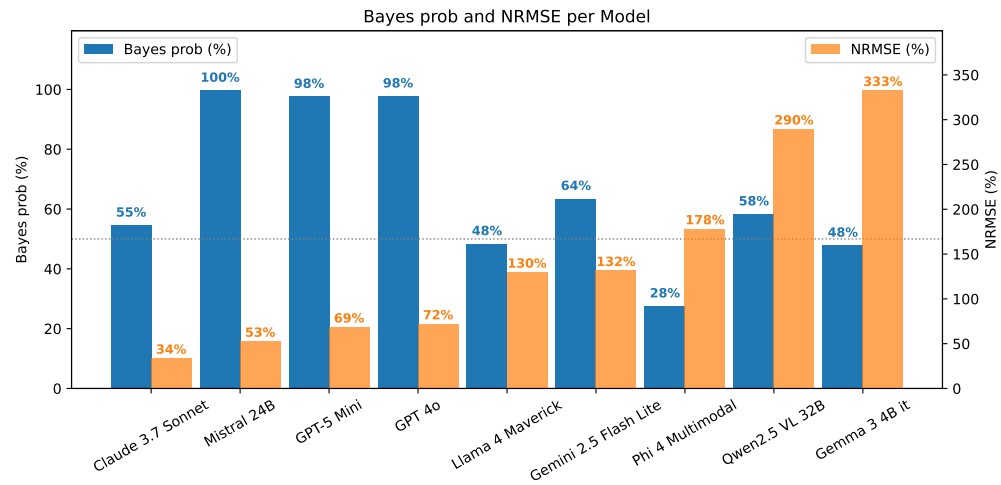

Figure 17: Subtitle duration estimation task. NRMSE and Bayes factor evidence.

For the Bayesian model we also added variants with an affine transform after the estimate is computed, to account for any potential gain biases. This is not needed for the linear models as it is captured by the gradient parameter.

Note that for all these variants, the additional parameters will penalise AIC and therefore help guard against artificial model evidence inflation by more complex models.

### A.13.1 LOGARITHMIC TRANSFORM

In some model variants, a logarithmic transform is applied to the stimulus or response space before fitting our behavioural models above. This is motivated by standard assumptions in psychophysics that humans internally represent magnitudes on a log scale.

Thus the transformed stimulus $x'_t$ from the raw input $x_t$ is

$$x'_t = \log(x_t + \epsilon),$$

with a small $\epsilon$ ensuring numerical stability. Log-transform variants are considered for both the linear and Bayesian observer models.

### A.13.2 AFFINE TRANSFORM

For Bayesian models, we additionally allow affine deviations of the posterior estimate, corresponding to a gain factor $g \in \mathbb{R}^+$ and an additive offset $\delta \in \mathbb{R}$. The raw posterior mean $\mu_t$ from the model estimate is transformed to $\tilde{\mu}_t$ as

$$\tilde{\mu}_t = g\,\mu_t + \delta.$$

The LLM response $y_t$ in these variants is generated as below, where $\sigma^2_{\text{dec}}$ is again a free parameter fitted during the model fitting stage:

$$y_t \sim \mathcal{N}(\tilde{\mu}_t,\, \sigma^2_{\text{dec}}).$$

This captures systematic deviations from the normative Bayesian solution, such as under- or over-weighting of evidence and constant response bias. Note that for linear models this is not required as it is already captured by the slope and offset parameters.

