# OpenReview forum: "Emergent Bayesian Behaviour and Optimal Cue Combination in LLMs"
_ICLR.cc/2026/Conference — Submitted to ICLR 2026_

### Official Review · Reviewer_sjdc · 2025-10-28

**Soundness:** 2
**Presentation:** 2
**Contribution:** 2
**Rating:** 2
**Confidence:** 3

**Summary:**

This paper investigates whether LLMs spontaneously exhibit human-like, near-optimal Bayesian strategies for multimodal integration without explicit instruction. The authors introduce BayesBench, a psychophysics-inspired benchmark featuring four magnitude estimation tasks across text and image modalities. By systematically manipulating noise and context, they evaluate nine LLMs on accuracy, cue-combination efficiency, and behavioral adaptation using a novel Bayesian Consistency Score (BCS). The findings reveal that high accuracy does not always imply efficient cue combination. However, more capable models, such as Llama-4 Maverick and Claude 3.7 Sonnet, demonstrate emergent behaviors that are consistent with Bayesian observer models, suggesting that principled uncertainty handling may be an emergent property of large-scale training.

**Strengths:**

1. Applying the classic psychophysics research framework to evaluate LLMs, this interdisciplinary method treats the models as "black-box observers" for systematic behavioral analysis, offering a novel perspective on the internal computational strategies they use to handle uncertainty.

2. BayesBench benchmark constructed by the authors not only evaluates task accuracy (NRMSE) but also measures cue integration efficiency and behavioral adaptation through Relative Error (RRE) and the innovative Bayesian Consistency Score (BCS).

**Weaknesses:**

1. The study's four magnitude estimation tasks are based on artificial and simplified scenarios such as line ratios as ASCII text, non-self-intersecting maze paths. These "toy" tasks fail to provide sufficient evidence that LLMs would apply the same Bayesian principles when processing the ambiguity and complexity inherent in real-world multimodal data.
2. While the research demonstrates that high-performance LLMs exhibit emergent Bayesian behavior, it fails to deeply analyze the origins of this emergence. For example, by not comparing models with different training objectives, it cannot validate its hypothesis that Bayesian strategies represent a universal solution derived from information-theoretic constraints.
3. The noise ablation study is narrowly focused, applying only Gaussian blur to the image modality. By neglecting to introduce noise to the text modality or explore other noise types, the study cannot determine whether the LLMs' strategies are consistent across different modalities and under diverse noise conditions.

**Questions:**

First, regarding generalizability, it might be valuable to explain how the behaviors observed in synthetic tasks could transfer to the ambiguity of real-world data—this thought arises partly from the limited scope of the noise study. Since optimal integration calls for flexibly re-weighting cues, showing how models perform with noisy text or conflicting cross-modal information could also help reinforce your claims.
Second, when it comes to the emergence of this behavior, your “universal solution” hypothesis is quite compelling, though additional direct evidence would enhance its persuasiveness.

---

> ### Author Response · Authors · 2025-11-24
> **Official Comment by Authors (Part I)**
>
> We thank the reviewer for the thoughtful comments and helpful framing. Below we respond point by point and indicate changes we will make in the revised manuscript.
>
> **Weakness 1: “Toy” magnitude-estimation tasks and real-world generalisability**
>
> We appreciate the reviewer drawing attention to the nature of the tasks in our experiments. We respectfully argue that the use of synthetic tasks is a prerequisite for a rigorous Bayesian analysis and follows the standard methods of human psychophysics (Ernst & Banks, 2002; Knill & Pouget, 2004)
>
> To assess whether an agent 1) acts in manner consistent with Bayesian principles and 2) perform efficient cue combination vs a Bayes-optimal combiner (the normative baseline), we need a tractable posterior P(state ∣ input)∝P(input∣state)P(state) for the observer, which requires knowing and controlling the generative process and the noise structure of the inputs.
>   - In naturalistic tasks (e.g., VQA on web images), noise distribution is unknown and uncontrollable. One cannot calculate the "optimal" weighting for the normative baseline
>   - In BayesBench, we explicitly control input visual noise and estimate LLM perception noise through repeated trials. These allow us to compute the ideal weighting for our normative baseline, forming a theoretical floor performance to measure against
>   - Manipulation of experimental conditions allow us to test if LLM adapts behavior in a Bayes-consistent way, acting as a probe for its strategy beyond performance
>
> BayesBench probes whether LLMs exhibit the characteristics (Bayes consistent adaptive behavior and noise-aware cue-combination) that are necessary for robust behaviour in more complex, real-world settings. This is directly analogous to how simple stimuli such as Gabor patches or random-dot kinematograms are used in neuroscience to uncover principles of perception that then inform our understanding of natural vision. While the tasks may appear simplified, the underlying computations required are the same.
>
> In the updated manuscript:
>   - Clarified in the Introduction motivations for the use of psychophysics-style tasks
>   - In the Discussion, clarified how the BayesBench framework could be extended to more naturalistic tasks (e.g. ambiguous captions, chart reading, conflicting text–image pairs).
>
> **Weakness 2: Limited analysis of the origins of “emergent” behaviour and the “universal solution” hypothesis**
>
> We agree that we currently do not have the experimental leverage to validate a universal, information-theoretic optimality hypothesis. Our model pool includes systems with different sizes, modalities, but we lack detailed access to their objectives and data, especially for proprietary models.
>
> While our goal is to find if there is evidence for a “universal solution”, we in fact find that model behavior is more nuanced and task specific. This was not emphasised strongly enough in our work and we will improve on this
>
> In the updated manuscript:
>   - Clarified in the Introduction and other paragraphs that relate to “universal” or “information-theoretic” explanations throughout the paper, clearly labelling them as hypotheses rather than established facts.
>   - Add a short paragraph in the Discussion noting that testing such hypotheses mechanistically would require controlled comparisons across models with known training objectives (e.g. variations in loss functions, data curricula, or explicit uncertainty objectives), which is beyond the scope of the current empirical study.
>   - Position in the Discussion section that BayesBench is providing the relevant behavioral measurements that could be used in exactly that kind of future, more mechanistic analysis

---

> ### Author Response · Authors · 2025-11-24
> **Official Comment by Authors (Part II)**
>
> **Weakness 3: Noise ablation only on images; lack of text/noise diversity**
>
> As discussed in our response to Weakness 1, BayesBench is explicitly designed around controlled manipulations of uncertainty, in the spirit of classical psychophysics, so that we can form a Bayes optimal or consistent baseline.
>
> Restricting the current noise ablation to the image modality via Gaussian blur was a deliberate step. Gaussian blur provides a well-controlled and objectively quantifiable manipulation of visual reliability. Holding the text description noise level fixed allows us to analyze the LLM's relative reliance on different modalities.
>
> We ran further analyses in the multi-modal line-length estimation task and found that LLMs generally reduce weighting on the image input when noise is introduced. In figure 6 (bottom right panel), the dark blue bars show the change in weighing on the image modality when LLMs are giving multimodal responses under image channel noise. A negative value indicates that the implied weighing has decreased, which makes sense when noise is only present in the image input. This illustrates the type of noise-reweighting BayesBench is designed to probe
>
> In the updated manuscript:
>   - Make explicit in the Methods section the design choice of Gaussian noise and the rationale of selective noise manipulation
>   - Added to the future-work section that future iteration may benefit from a richer set of noise manipulation
>
> **Addressing specific questions:**
>
> **Q1 & 3: Generalisability from synthetic tasks and broader noise/conflict settings**
>
> We frame our synthetic tasks as idealized probes of a more general set of capabilities: we are attempting to use simple scalar magnitude estimation to probe model adaptation when experimental environment changes (corresponding to the various ablations we used). Synthetic tasks allow us to easily manipulate their inputs and have minimal risk of models having seen them in pretraining. We believe many real-world problems can be decomposed into such local operations, even if the full scene is far more complex, and recording performance in these examples represent the key first step of more general behavioral studies. In our study, we find that some current LLMs already exhibit Bayes-consistent adaptations in these basic settings.
>
> That said, we fully agree that demonstrating similar behaviour in more realistic, ambiguous scenarios would be valuable. Our present work establishes the benchmark and metrics; extending them to richer stimuli and more diverse noise types will be a key next step.
>
> In the updated manuscript, we added in the Discussion section additional description connecting the synthetic tasks to classes of real-world problems and explaining the limits of this analogy.
>
> **Q2: Universal solution hypothesis as an emergent property**
>
> We reiterate that our “universal solution” framing is an initial hypothesis we wish to explore, and our current experiments show that while different high-capability models, trained independently, seem to display more Bayes consistency as they become more accurate, different models often display context specific behavior and therefore it is not clear from our experiments that a universal solution exists currently.
>
> Regarding an analysis of the mechanistic origin of these behaviors, we agree that this will be a really valuable next step. For the time being, as a number of our models are proprietary with undisclosed training objectives, this is beyond the resources available in our work.
>
> In the updated manuscript, we expanded the future-work section to (i) emphasise noisy/conflicting cross-modal conditions and (ii) identify the mechanistic origins of these behaviors by analyzing controlled training comparisons. We will also reiterate that the “universal solution” is a hypothesis of ours that is not yet clear from the current class of models we studied.

---

### Official Review · Reviewer_sQto · 2025-10-29

**Soundness:** 2
**Presentation:** 3
**Contribution:** 2
**Rating:** 4
**Confidence:** 4

**Summary:**

This paper investigates whether LLMs implicitly develop Bayesian computational strategies for processing uncertainty, similar to those observed in human perception. The authors introduce BayesBench, a novel benchmark inspired by classic psychophysics, which consists of four magnitude estimation tasks (length, location, distance, duration) across text and image modalities. To evaluate model behavior, they propose a new metric, the Bayesian Consistency Score (BCS), alongside standard accuracy (NRMSE) and cue-combination efficiency (RRE) metrics. The BCS is designed to measure whether a model's behavior shifts in a Bayes-consistent direction in response to controlled ablations (e.g., noise, context changes). The key findings are :(1) more capable models, such as GPT-5 Mini, Llama-4 Maverick, and Claude 3.7 Sonnet, often exhibit Bayes-consistent adaptations, and (2) high task accuracy (as seen in GPT-5 Mini) does not necessarily correlate with efficient or optimal multimodal cue combination.

**Strengths:**

The paper's primary strength lies in its originality and interdisciplinary approach. Applying the rigorous, time-tested paradigm of psychophysics to probe the implicit computational strategies of LLMs is a highly novel and insightful direction, moving beyond standard accuracy-based evaluations.

The methodological contribution, BayesBench, is solid. It provides a controllable and reproducible framework for testing how models handle uncertainty. The use of controlled ablations (noise, context, steering) is a systematic way to probe behavioral shifts.

Furthermore, the introduction of the Bayesian Consistency Score (BCS) is a clever conceptual advance. By focusing on the direction of behavioral change in response to perturbations (e.g., how the prior weight shifts) rather than just the static goodness-of-fit to a single Bayesian model, the BCS attempts to capture a more principled, adaptive strategy.

Finally, the paper delivers a valuable and nuanced finding: the decoupling of task accuracy from cue-combination efficiency. The observation that a highly accurate model like GPT-5 Mini can be an inefficient multimodal integrator is an important insight for the future design and evaluation of robust multimodal systems.

**Weaknesses:**

Despite the novel premise, the paper's central claim—that LLMs exhibit "emergent Bayesian behaviour"—is not adequately supported, as it rests on several questionable assumptions and interpretations.

The most significant weakness is the conflation of "regression-to-the-mean" with Bayesian inference. The primary evidence for Bayesian processing is the regression effect shown in Figure 1, where estimates are biased toward the center of the stimulus range. While this pattern is consistent with Bayesian integration (a prior pulling the likelihood), it is not sufficient evidence. Many simpler, non-Bayesian heuristics, such as anchoring on the mean of the current session's stimuli, could produce an identical pattern. The paper fails to test or rule out these more parsimonious alternative explanations.

This issue is compounded by a major confounding variable: in-context learning (ICL). The experimental setup explicitly provides a "rolling context" of previous trials and responses, which the authors state is the "basis of the emergence of Bayesian consistent behaviour". This design makes it impossible to distinguish between a genuinely "emergent" computational strategy and the model simply performing standard ICL on the provided history. The "Static Bayesian observer" or "Sequential Bayesian observer" might just be effective descriptions of a model learning the stimulus statistics from the prompt context, rather than reflecting a fundamental, internalized mechanism for handling uncertainty.

Furthermore, some of the paper's own findings contradict its central narrative. The discovery that Llama-4 Maverick outperforms the Bayes-optimal linear fusion model is a striking result. However, attributing this to a "sophisticated non-linear integration strategy" undermines the claim that models are converging on the classic Bayesian observer principles (which are typically based on linear-Gaussian assumptions, i.e., BLUE). If the best-performing model uses a strategy that is non-linear and better than the Bayesian benchmark used, it suggests the benchmark itself is an incomplete or incorrect model for what these LLMs are doing.

Finally, the claim of a "universal" strategy is weakened by task-specific inconsistencies. In the Maze Distance task, GPT-5 Mini achieves "near-perfect text performance", which is clearly a product of explicit reasoning, not implicit perception. If the model uses explicit reasoning for the text modality while (supposedly) using implicit Bayesian strategies for the image modality, it suggests an opportunistic, task-dependent application of different tools, not the emergence of a unified, human-like perceptual processing strategy.

**Questions:**

1.	How can the authors definitively distinguish "Bayesian computation" from simpler heuristics like "regression-to-the-session-mean"? Given that the session mean is easily discoverable via ICL from the "rolling context", what evidence shows this is more than just anchoring? Were any non-Bayesian control models (e.g., a simple anchoring-and-adjustment model) fit to the data?
2.	The "rolling context" seems critical for all sequential effects. What happens if this context is removed (e.g., by running trials in isolation or resetting the state)? If the Bayesian signatures disappear, wouldn't this confirm that the observed behavior is an artifact of ICL rather than an "emergent" property of the model's perceptual system?
3.	Regarding the Llama-4 Maverick finding: If the best-performing model (non-linear) surpasses the Bayes-optimal linear model (BLUE), does this not suggest that the linear-Gaussian Bayesian observer is the wrong benchmark? How does this finding support the conclusion that LLMs are converging on these specific Bayesian principles?
4.	In the maze task, the model appears to mix explicit reasoning (text) with implicit perception (image). How does this fit the narrative of an "emergent Bayesian" strategy? Does this not imply that the model lacks a unified strategy and simply defaults to different mechanisms based on modality and task difficulty?
5.	Could the authors further justify the design of the BCS? Why sum the binary signs of the change (e.g., plus or minus one) rather than a more nuanced metric, such as the correlation between the magnitude of the intervention (e.g., level of noise) and the magnitude of the shift in the prior weight? The current metric seems to lose a significant amount of information.

---

> ### Author Response · Authors · 2025-11-24
> **Official Comment by Authors (Part I)**
>
> We thank the reviewer for the careful and constructive comments. Below we respond point by point and indicate changes we made in the revised manuscript.
>
> **Weakness 1: Regression-to-the-mean vs Bayesian inference; lack of non-Bayesian controls**
>
> We fully agree that regression to the mean alone is not sufficient evidence of Bayesian computation as other simple heuristics (e.g. regression-to-session-mean) can produce similar static patterns. Our intention is not to equate regression with being Bayesian, but to use a Bayesian observer model as a descriptive probe and then focus on how its parameters change under controlled manipulations. This is why we introduce the Bayesian Consistency Score (BCS).
>
> All our tasks are identity-mappings (LLM is given a stimulus and asked to produce its estimate. The perfect estimator will be an input-output identity mapper). For such tasks, a broad class of non-Bayesian heuristics can be expressed as monotone transforms plus noise. Our unconstrained linear observer (plus a log-transform variant) is designed to cover exactly this family: it can regress to any anchor in the range, shrink estimates towards a session mean, apply gain distortions, etc., without any probabilistic semantics. The “Bayes factor evidence” we report is based on AIC comparisons across this model set, not on regression patterns alone.
>
> While goodness-of-fit is relevant, we see the more relevant metric as the Bayesian Consistency Score (BCS), which measures the directional parameter changes across ablations (noise, context, steering). BCS asks whether the model adapts its behavior in a way consistent with Bayesian theory when we change the experimental conditions.
>
> We implemented in the updated manuscript:
>   - Clarify in section 3.2 that regression to the mean is compatible with but not definitive proof of Bayesian inference, and that we therefore focus on changes in parameters in response to changes in experimental environment (BCS) rather than on regression effects alone to detect Bayesian consistent behavior.
>   - Clarify in the description of the unconstrained linear baseline in section 3.2 that it already captures a “regression-to-anchor” type heuristic, which we treat as a non-Bayesian comparator.
>
> **Weakness 2: In-context learning as a confound; “emergent” vs prompt-level learning**
>
> We see Bayesian learning and ICL as complementary: from a computation point of view, a Bayesian learner is an agent that forms expectations after being exposed to relevant context and uses that to refine its estimates.
>
> Our experiments are explicitly designed to probe how models use the provided rolling context of previous trials and feedback. ICL is a key mechanism through which Bayesian behavior can arise. Through our BCS score we find that different model indeed perform Bayes consistent behavior to different extent. A related point here is the steering ablations we performed. When we provide to the model a numerical prior that is intentionally biased, we indeed see worse estimation performance, showing that models are indeed taking into account in-context information when making estimation.
>
> In the updated manuscript, we clarified in section 3.5 that our results characterise how models behavior changes when environment changes.

---

> ### Author Response · Authors · 2025-11-24
> **Official Comment by Authors (Part II)**
>
> **Weakness 3: Llama-4 Maverick outperforming the Bayes-optimal linear fusion model**
>
> We appreciate that this is a surprising result and wish to clarify the rationale and purpose of the RRE metric. This metric is used to measure cue-combination performance against a reasonable and widely used Bayesian baseline rather than to measure behavioral fit.
>
> The Bayes-optimal linear combiner is a normative baseline under a Gaussian‐noise assumption. This is used in many biological studies including human perception. In reality, LLM are not constrained to use linear combination of unimodal responses, nor to have Gaussian noise. Performance of this normative model represents the reasonable level of a rational model. The RRE benchmark compares measured LLMs' accuracy against this.
>
> When Llama-4 Maverick surpasses this baseline, it is a significant finding as it implies that it can leverage non-linear relationships not assumed under our normative baseline - ie a strategy that will outperform how biological systems, including humans, typically integrate information. As shown in figure 6, Llama 4's multimodal response is better fitted using a non-linear random forest model against its unimodal responses vs linear variants.
>
> However, Llama-4 Maverick's performance level is not generally matched by others. Our benchmark thus serves as a valuable target: for future models, we aim to have their multimodal performance be at least comparable with our Bayesian baseline, as this is the performance expected of a rational combiner
>
> We implemented in the updated manuscript:
>   - Clarified in sec 3.5 that the cue combination benchmark is meant to measure efficiency against a reasonable baseline, rather than prove computation
>   - Add in sec 5.2 when we discuss Llama-4's outperformance the implications on known information combination principles in biological systems
>   - Add in sec 6 that for other less performant LLM's this serves as a valuable target
>
> **Weakness 4: Task-dependent strategies (e.g. reasoning in Maze Distance) vs a “universal” strategy**
>
> We agree that our study indeed indicates different tasks invite different strategies from LLMs and this is an important observation. For example, GPT-5 Mini’s near-perfect performance in the text-based Maze Distance is a result of explicit reasoning rather than implicit perception when we analyze its responses.
>
> While our original question was to ask if there exists a single, unified computation strategy, we in fact found that LLMs' behaviours are more nuanced and context dependent. BayesBench is intended to reveal where Bayes-consistent patterns appear, and on average how well models stand against each other in terms of task accuracy and in terms of cue-combination efficiency.
>
> We implemented in the updated manuscript:
>   - Clarify any language suggesting we found a “universal” emergent Bayesian strategy - this was instead a question at the beginning of the research.
>   - Frame our contributions as exposing where and when Bayes-consistent behaviour arises across tasks and modalities.
>   - Use the Maze task as an example of opportunistic strategy selection (reasoning vs perceptual-like estimation), which BayesBench helps to expose.
>
> **Addressing specific questions**
>
> **Q1. Distinguishing Bayesian computation from regression-to-session-mean / anchoring**
>
> Please see further details on weakness 1 above. Our unconstrained linear baseline effectively implements a "regression-to-anchor point" heuristic: it can place the anchor anywhere in the observed range and shrink estimates towards it without any probabilistic semantics. This together with its log-transform variant cover a wide range of relevant non-probabilistic models for our identity mapping tasks.
>
> We agree that simple model evidence alone is an insufficient proof of Bayesian computation. Therefore in our analysis, the role of the Bayesian observer models is instead to provide a structure on which behaviour changes can be measured across ablations. Our BCS metric is computed on parameter changes (e.g. does the inferred prior weight increase when we add noise?) across conditions rather than static goodness-of-fit alone. We will make this limitation and intention clear; this is why we consistently talk about “Bayes-consistent” behaviour rather than a definitive mechanistic account. Overall the BayesBench score considers task performance, cue combination efficiency and Bayes consistent behavior, rather than just the goodness of fit to behavioral models.
>
> We implemented in the updated manuscript:
>   - Added in sec 3.2 clarification of how the linear model acts as an anchoring-type heuristic and make clear that we use the Bayesian observer as a descriptive probe in our BCS and not relying on goodness-of-fit alone as proof of Bayesianness.
>   - Add to the Discussion that more specialised heuristic models (e.g. explicit anchor-and-adjust rules) would be an interesting comparison for future work where experiment covers more intricate tasks.

---

> ### Author Response · Authors · 2025-11-24
> **Official Comment by Authors (Part III)**
>
> **Q2. Role of rolling context and ICL**
>
> Please see further details in the section on weakness 2. Sequential effects in our experiments indeed depend on having previous trials and feedback available in the LLM's context. If we were to remove the rolling context entirely—so that each trial is presented in isolation—we would not expect to see the same pattern of history-dependent behaviour. This is similar to studies in humans, where a number of examples have to be given before any form of prior can take shape.
>
> We do not see this as an issue, as any Bayesian behavior that can arise must involve leveraging information present in its context, giving it a history to base estimation on. The main question is: given the conditions for forming a prior (enabled by rolling context), does the model reweight prior vs current information consistent with Bayesian predictions under ablations? This is captured by our BCS metric.
>
> We implemented in the updated manuscript:
>   - Clarified in the Methods and Experimental Setup section that our setup intentionally leverages rolling context and that our claims are about Bayes-consistent use of in-context information when conditions change.
>
> **Q3. Llama-4 Maverick surpassing the Bayes-optimal linear model**
>
> Please see segment on weakness 3 above for further details. Our linear Bayesian fusion model sets a simple normative baseline for LLM multimodal performance. Llama-4 Maverick surpassing it suggests that the model can exploit additional structure beyond the scalar Gaussian assumption. This is significant as it indicates a mechanism that would outperform how known biological systems work. This is confirmed in figure 6, where its multimodal responses are better fitted against unimodal responses with a non-linear random forest than with linear variants.
>
> The RRE metric in BayesBench is not used to infer the exact computation principle but to compare performance against a reasonable Bayesian combiner baseline.
>
> We implemented in the updated manuscript:
>   - Clarified in section 3 the assumptions behind our RRE benchmark and clarify that we are using this as a normative baseline, rather than behavioral fit, to quantify LLM's cue combination efficiency.
>
> **Q4. Mixing explicit reasoning and implicit perception in the Maze task**
>
> We agree that the Maze task highlights that models mix different mechanisms: explicit reasoning on text inputs and more perceptual-like estimation on images. This is compatible with our aims: BayesBench is designed to reveal this kind of heterogeneity. As Bayesian computation is a consistent way to deal with uncertainty, revealing instances where model do not adapt in a Bayes-consistent manner can inform future model development.
>
> We implemented in the updated manuscript:
>   - Clarified in the Introduction and Discussion section that while we set out to find if there are "universal" computations that are intrinsically Bayesian, our experiments reveal that model behavior is more nuanced and multifaceted.
>   - Clarified in the Results section in the Maze results noting explicitly that text performance by GPT-5 mini reflects explicit reasoning, and that this underscores the task-dependent nature of LLM's strategies.
>   - Clarified in the Discussion that our framework is useful because it can show such differences across modalities within the same task.
>
> **Q5. Design of the Bayesian Consistency Score (BCS)**
>
> We designed BCS as a deliberately simple, scale-free summary of directional consistency. We use the signs of parameter changes rather than magnitudes for these reasons:
>   - The absolute magnitude of a parameter shift depends on the internal scales and parameterisation of the LLM and on the baseline noise level of each LLM. Taking only the sign of the change makes BCS robust to these differences and to rescalings of parameters.
>   - Limited ablation levels. In each task we use a small number of ablation conditions (e.g. low/high noise, with/without prompt steering). With so few levels, estimating reliable correlations between intervention magnitude and parameter change can be unreliable and be dominated by model-specific scaling factors.
>
> We implemented in the updated manuscript:
>   - Clarified the design goals of BCS (robustness, scale-free directional test) in the Methods section
>   - Add that for future work, a more diverse set of experiments may benefit from a version of the BCS that takes into account further details around magnitude of parameter changes

---

### Official Review · Reviewer_3KTP · 2025-10-30

**Soundness:** 3
**Presentation:** 2
**Contribution:** 3
**Rating:** 4
**Confidence:** 4

**Summary:**

This manuscript studies the behavior of large language models (LLMs) under four behavioral tasks (marker location estimation, line ratio estimation, maze distance estimation, duration estimation). These tasks were previously used to study human behavior. Bayesian models have been developed to explain the behavior of human in these tasks.  The paper assessed the extent by which the behavior of a number of LLMs resembles Bayesian computations. Empirical observations on the performance of these models were reported in this study.

**Strengths:**

The idea of testing LLMs in a range of commonly used psychophysical and behavioral tasks is interesting.

The study tested four tasks on a number of large language models. Three of them were multi-modal, so it was possible to assess whether optimal Bayesian cue combination strategies were used in LLMs for these tasks.

The authors considers several models of the observer’s behavior, i.e., linear observer, static Bayesian observer, Kalman filter.

The authors developed several metrics to evaluate the behavior of LLMs.

The behavioral benchmark (BayesBench) will be shared with the community and may be of interest to some other researchers as well.

**Weaknesses:**

— While several tasks were used to test several variations of LLMs, there is not a major insight learned from the study.

— It was not clear how to interpret the finding that some LLMs were able to better integrate the information from the two modalities. Does this have something to do with how these models were trained (differently)?

— The writing needs improvement. In various places of the paper, the interpretations of the prior literature were not accurate. I would like to suggest a careful check of the accuracy of the reference to the literature.

— It was unclear how the main results depend on the details of the prompts for LLMs. For example, if the LLMs were instructed to ignore their responses to the previous trials, would the sequential effect (akin to Kalman filter still persist)?

**Questions:**

What do the dots with different sizes represent in Fig. 1A?

Given the task setting, what are the optimal strategies? For example, would a Kalman filter be optimal? The authors fit three classes of models ( linear observer, static Bayesian observer, Kalman filer ) to the behavior data, however, the Bayesian optimal was not established. So it is difficult to assess how close the LLMs perform relative to the optimal strategy.
For experiments with three different ranges, what if trials were interleaved between the three ranges? Would that change the behavior of the model?

Form Fig. 4, GPT-Mini’s responses did not seem to exhibit a regression toward the mean. Can the authors clarify?

Section 5.3 says “While Gemma 3 4B and Phi 4 Multimodal achieved higher BCS than expected, they are also the least accurate group of models. ” This is somewhat confusing. In what sense, it achieves higher BCS than “expected”?

---

> ### Author Response · Authors · 2025-11-24
> **Official Comment by Authors (Part I)**
>
> We thank the reviewer for the thoughtful and detailed comments. We detail below our responses and indicate changes we made in the revised manuscript.
>
> **Weakness 1: “No major insight” from the study**
>
> Our paper is positioned as a first, systematic step in bringing classic human psychophysics paradigms to LLMs and in establishing a set of behaviourally grounded, Bayesian-relevant metrics
>
> One key insight is that accuracy and robustness dissociates. High-performing models like GPT-5 Mini achieve near-perfect accuracy in some tasks yet fail to optimally perform cue integration compared to a simple Bayes optimal combiner. Conversely, some smaller models (e.g., Gemma 3, Phi 4) exhibit relatively low raw accuracy but comparatively high Bayesian Consistency Scores (BCS), indicating that their pattern of adaptation to noise is closer to the ideal observer than their overall error rate might suggest.
>
> This dissociation has direct implications in practice: in real-world deployments where input quality varies and robustness matters at least as much as average accuracy, leaderboards based solely on accuracy may favour brittle systems. BayesBench provides a complementary layer of diagnostic via BCS and Bayes-RRE. These can reveal robustness properties invisible to accuracy alone.
>
> Other notable observations and contributions:
>   - Psychophysical identity-mapping tasks transfer to LLMs. Magnitude-estimation tasks reveal familiar human-like patterns in some models.
>   - BayesBench jointly probes capability and strategy. NRMSE captures task accuracy, Bayes-RRE quantifies multimodal cue-combination efficiency relative to an Bayes-optimal linear combiner, and BCS measures changes in behaviour under controlled noise manipulations. This combined metric goes beyond simply accuracy or goodness-of-fit.
>   - The relationship between model scale, accuracy, and Bayesian behaviour is nuanced. While stronger models tend to achieve higher accuracy and better BCS on average, there are exceptions in both directions (high-accuracy but cue-inefficient models; lower-accuracy but relatively Bayes-consistent models).
>
> We see this paper as identifying potentially critical gaps in understanding and benchmarking LLM behavior, and laying the ground work for further analyses by introducing an alternative benchmark and metric suite.
>
> We have revised the manuscript accordingly and reflected the following changes:
>   - Made sure to clarify these concepts in Abstract and Introduction (Sec 1). We have also elaborated further in Cue Combination (Sec 5.2) and Discussion (Sec 6) these conceptual takeaways (capability vs consistency, cue combination vs general accuracy, complementary metrics vs accuracy alone) so the main insights are easier to see.
>   - Add in Discussion that our work forms a first step and that BayesBench is intended as an infrastructure for deeper follow-up studies.
>
> **Weakness 2: Interpretation of “better integration” across modalities**
>
> We agree that we should be more explicit about what can be concluded when some models integrate cues more effectively. Our empirical claim is purely behavioral. We see some LLMs psroduce combined-cue estimates whose accuracy is closer to a Bayes-optimal linear combiner constructed from their own unimodal responses. This is quantified by Bayes-RRE. Higher Bayes-RRE indicates that, given how the model performs in the single-cue conditions, it leverages the two cues together in a way that is closer to the normative baseline. Better integration means a performance closer to this normative baseline.
>
> We do not have full visibility into the training recipes of proprietary models (e.g. GPT, Claude, Gemini), so we do not attempt a causal attribution at the architectural level at this stage. However, we still see notable patterns. For example, that multimodal models with strong vision training such as Llama 4 Maverick exhibit strong cue combination efficiency, while smaller models (e.g. Gemma 3 4B, Phi 4 Multimodal) sometimes show lower cue combination efficiency. We will emphasise that these are descriptive observations provided by our diagnostic metrics (Bayes-RRE + BCS).
>
> We have revised the manuscript accordingly and reflected the following changes:
>   - Clarified in Discussion (Sec 6), explicitly stating that we only make behavioral observations and do not claim a detailed causal explanation from training data or architecture.
>   - Clarify that Bayes-RRE (Sec 3.3 and Sec 5.2) is defined relative to each model’s own unimodal responses. This metric is used as a normative accuracy reference, where we are asking how well a model utilizes its own unimodal responses, in comparison to a simple linear but Bayes-ideal combiner.

---

> ### Author Response · Authors · 2025-11-24
> **Official Comment by Authors (Part II)**
>
> **Weakness 3: Writing and accuracy of references to prior literature**
>
> We appreciate this comment and agree that we should tighten both the writing and the way we describe prior work. We will:
>   - Carefully review all references to prior psychophysics and Bayesian modelling work, cross-checking claims against the original papers, and soften or correct any statements that over-generalise.
>   - Improve clarity in the Introduction and Related Work by tightening sentences and adding citations where missing.
>
> **Weakness 4: Dependence of results on prompt details**
>
> We agree that prompt dependence is important and appreciate the reviewer drawing attention to this. In fact, we already explored prompt variants as part of our steering ablations:
>   - 1) Verbal steer: we add in the prompt that models should "behave like a Bayesian observer and take into account prior and likelihood in its predictions"
>   - 2) Numerical steer: we add in the prompt additional numerical information "For 10 previous observations, the values were observed to lie in the range of [x] to [y]"
>
> We found that numerical ablation has a bigger impact on model behavior than verbal steer. When a numerical range is provided,  models generally show reduced sequential behavioral evidence. We ran further experiments with a biased numerical range in the prompt (ie, giving to the model a range that is deliberately off) and model prediction error increased. Indicating that models are influenced by the explicit range and do not fully override it with online evidence
>
> In the updated manuscript, we have added additional experimental results:
>   - error decomposition of different models under the steering ablations for the length ratio task. Under the biased numerical steer (triangles), RMSE is generally higher compared to the base case (circle) - see panel A of figure 6
>
>   - sequential evidence under steering ablation. Models generally show lower evidence of sequential behavior when under numerical steer (see panel B of figure 6)
>
> In the updated manuscript
>   - Clarified the prompt ablations (Sec 3.1 under Steering ablation) and provide the exact prompt variants in an appendix 3.1).
>
> **Addressing specific questions**
>
> **Q1. What do the dots with different sizes represent in Fig. 1A?**
>
> We apologise for the lack of clarity. In Fig. 1A, the smaller dots represent individual responses, and the larger dots represent the corresponding mean responses within each bin (binned by trial number). We will clarify this in the document
>
> **Q2. Optimal strategies and interleaving of ranges**
>
> In our tasks, the optimal strategy depends on each LLM’s perceptual capability. A model with very low noise can behave almost like an identity mapper and be close to optimal; while a noisier model will benefit more from regressing towards its prior. What is consistent across models, and what BayesBench is also designed to measure, is how behaviour changes when condition changes. For example, when we increase noise, alter context, or manipulate range, a fitted Bayesian observer should show parameter shifts in the Bayes-predicted directions (e.g. increased prior weight when sensory noise increases). The BCS score captures exactly these directional changes across ablations; this cross-condition structure is a main focus of our analysis
>
> In the updated manuscript:
>   - Clarified in Methods (sec 3.5) that while per-condition optimal strategies depend on each model’s noise level, our primary interest is in the relative parameter changes across conditions, as measured by BCS.
>
> **Q3. Regression to the mean in GPT-5 Mini (Fig. 4)**
>
> GPT-5 Mini’s responses show relatively little regression to the mean because its point estimates are already extremely accurate which indicates that it is likely able to identify the input without reliance on the prior (ie, its likelihood function is very reliable). In a Bayesian account, when measurement noise is very small, the posterior mean can be essentially dominated by the likelihood and thus regression effects become minimal. In that limit, Bayesian and “perfect linear” behavior become indistinguishable. This is precisely why we introduce BCS: by focusing on changes under ablations, BCS can reveal Bayesian-like structure even when static effects like regression slopes are small.
>
> In the updated manuscript, we added a short explanation in above Fig. 4 that models with very low noise (high accuracy) will naturally show weaker regression-to-the-mean in a Bayesian account.

---

> ### Author Response · Authors · 2025-11-24
> **Official Comment by Authors (Part III)**
>
> **Q4. “Higher BCS than expected” for Gemma 3 4B and Phi 4 Multimodal**
>
> Our phrase “higher BCS than expected” was intended to capture the following empirical pattern: these smaller, less accurate models have relatively high Bayesian Consistency Scores given their overall error.
>
> We initially anticipated that BCS would track accuracy (i.e. more accurate models would be more Bayes-consistent), but we observe that some smaller, less accurate models (Gemma 3 4B, Phi 4 Multimodal) have relatively high Bayesian Consistency Scores compared to what one might predict from their task accuracy alone.
>
> In the updated manuscript we clarified in Sec. 5.3 that these models are less accurate but relatively Bayes-consistent given their capability, ie that their BCS was higher than we would predict based solely on their error levels

---

### Official Review · Reviewer_d9eK · 2025-11-01

**Soundness:** 3
**Presentation:** 2
**Contribution:** 3
**Rating:** 4
**Confidence:** 4

**Summary:**

This paper introduces BayesBench, a psychophysics-inspired benchmark for evaluating whether large language models (LLMs) perform optimal multimodal integration consistent with Bayesian principles. The authors adopt classic psychophysics paradigms to assess nine LLMs across four magnitude-estimation tasks (location, ratio, distance, and duration) in both text and image modalities. A central challenge in this investigation is manipulating the precision of different information sources to test the adaptivity of LLM behavior. To address this, the authors use three methods—prompt steering, noise injection, and context manipulation—to systematically vary the precision of prior knowledge and observational evidence. Beyond standard accuracy metrics, they introduce a Bayesian Consistency Score to detect Bayes-consistent behavioral patterns even when performance saturates. Their results reveal that high accuracy does not necessarily imply efficient cue combination, though several accurate models exhibit Bayes-consistent adaptations, suggesting emergent principled uncertainty handling.

**Strengths:**

The paper makes several notable contributions. First, the authors develop a rigorous pipeline enabling controlled ablation of noise, context, and instruction prompts, which provides systematic leverage for probing LLM computational strategies. Second, the choice of perceptual tasks is well-motivated, as these tasks are less susceptible to contamination from previously acquired statistical rules or memorized patterns compared to knowledge-based tasks. Third, the evaluation framework is comprehensive, incorporating tense and multifaceted measures that go beyond simple accuracy to probe behavioral consistency with Bayesian principles.

**Weaknesses:**

**1. Confounding between capability and decision strategy.** The relationship between overall RMSE and Bayes factor evidence (Figure 7, line 376) raises fundamental questions about what is actually being measured. Since overall RMSE is partly influenced by Bayesian inference itself, the observed negative correlation may simply reflect circular reasoning rather than revealing genuine Bayesian tendencies specific to particular LLMs. A critical issue in establishing valid tests of Bayesian inference is separating an agent's perceptual or representational capabilities from its decision-making strategies. Without this separation, it remains unclear whether models with lower RMSE exhibit more Bayes-consistent behavior because they implement Bayesian computations or simply because they have superior underlying capabilities that coincidentally align with Bayesian predictions.

**2. Limited evidence for benchmark generalizability.** Before the impact of this benchmark can be properly evaluated, critical information about cross-task and cross-modality consistency is needed. If a specific LLM shows substantially different Bayesian consistency across tasks and modalities, establishing a general benchmark for Bayesian inference becomes questionable. The paper would benefit from reporting consistency metrics across conditions and discussing whether the observed patterns reflect stable computational principles or task-specific behaviors. While the authors acknowledge the need for extension, the current scope may be insufficient to support broad claims about LLM Bayesian reasoning. Expanding to additional diverse tasks would strengthen confidence in the benchmark's validity.

**3. Weak experimental design for testing Bayesian inference.** The use of uniform priors constitutes a relatively weak test of Bayesian reasoning. Under uniform priors, simple heuristic algorithms (such as averaging cues with task-dependent weights) can produce patterns resembling Bayesian integration without implementing genuine probabilistic inference. Stronger tests would involve informative priors with varying shapes or hierarchical structures that more clearly distinguish Bayesian from non-Bayesian strategies.

**4. Insufficient evaluation of alternative explanations.** The paper focuses almost exclusively on Bayesian frameworks without thoroughly evaluating non-Bayesian alternatives—a concern frequently raised in human psychophysics literature. Simple heuristics (e.g., reliability-weighted averaging, salience-based selection, or context-dependent switching rules) could potentially explain observed patterns without invoking Bayesian principles. Without systematic comparison against reasonable non-Bayesian computational models, the interpretation that LLMs exhibit Bayesian reasoning may be incomplete or potentially misleading. The authors should either formally test alternative models or provide stronger behavioral signatures that uniquely implicate Bayesian computation.

**Questions:**

### Q1: Separating Capability from Decision Strategy
How do you disentangle an LLM's perceptual/representational capabilities from its decision-making strategy? Developing theoretical or experimental measures for the perceptual/representational capabilities in question would help to answer this question.
### Q2: Cross-Task and Cross-Modality Consistency
How consistent is each LLM's Bayesian behavior across the four tasks and two modalities? Does a model exhibiting Bayes-consistent behavior on one task show similar patterns on others?
Please report within-model consistency statistics, such as correlation of Bayesian Consistency Scores across tasks and modalities. If consistency is low, discuss whether this indicates task-specific behaviors rather than general Bayesian computational principles.
### Q3: Stronger Tests of Bayesian Inference
Have you considered experimental designs with non-uniform, informative priors? Under uniform priors, weighted averaging heuristics can closely mimic Bayesian predictions.
Please consider to include experimental conditions with informative priors, such as skewed or bimodal distributions, where Bayesian and heuristic predictions diverge. Alternatively, provide theoretical justification for why uniform priors are sufficient or acknowledge this as a limitation.
### Q4: Evaluation of Non-Bayesian Alternative Models
What specific non-Bayesian computational models have you tested? Can simpler heuristics explain the observed patterns? Fitting a larger set of alternative models, especially those not in the Bayesian framework, and reporting the results of model comparison would help to address this question. Please demonstrate that specific key findings cannot be captured by plausible simple heuristics.
### Q5: Behavioral Signatures Beyond Regression to Mean
Beyond regression-to-mean effects, what specific behavioral patterns uniquely implicate Bayesian computation? For sequential models, do you observe learning curves or posterior sharpening with accumulated evidence?

### Additional Comments
* Human subject studies require an ethics statement. Was this work approved by an Institutional Review Board? I may have missed something but I did not find any ethics statement in the current manuscript.
* For practical deployment of BayesBench, guidance is needed on the minimum number of trials necessary to establish reliable estimates of Bayesian consistency.
* Beyond regression-to-mean patterns, what are the distinctive behavioral hallmarks of Bayesian inference in your data? For sequential Bayesian models specifically, does deviation from ground truth systematically change with accumulated experience?
* Figure 3 shows GPT-5 Mini's responses against sequential Bayes predictions. Including ground truth values as reference would improve interpretability.
* How were data handled when less capable LLMs failed to follow instructions properly?
* Figure 4 would benefit from an identity line for reference.
* Line 215: Please clarify the distinction between "behavioral modeling" and "cue-combination modeling".
* In metric definitions, do subscripts "LLM" and "model" refer to the same entity? Please standardize terminology.
* Given GPT-5's near-perfect performance, is it possible the model is secretly using external tools?
* "Bayesian factor evidence" appears in Figure 7 and Appendix A7 but lacks clear definition in the main text. Please provide explicit explanation where first introduced.

---

> ### Author Response · Authors · 2025-11-24
> **Official Comment by Authors (Part I)**
>
> We thank the reviewer for the thoughtful and detailed comments. We detail below our responses and indicate changes we will make in the revised manuscript.
>
> **Weakness 1: Confounding between capability and decision strategy**
>
> We thank the reviewer for drawing attention to this aspect. Our experimental design in fact targets exactly this separation, and we will make this much more explicit.
>
> All four tasks in BayesBench are identity-mapping tasks: the model is instructed to report the magnitude of a presented stimulus, analogous to studies in classic human psychophysics. We separate capability and decision strategy as:
>   - Capability as how precisely the model reproduces the stimulus in a base condition. This is quantified by RMSE/NRMSE (and by Bayes-RRE in cue combination efficiency, see further details below); and
>   - Decision strategy as how models' behavior changes when we manipulate test environment. This is quantified by the Bayesian Consistency Score (BCS), which looks at changes of fitted Bayesian parameters across ablations. This is distinct from goodness-of-fit measure in a single condition.
>
> The relationship between NRMSE and Bayes-factor evidence is an empirical observation rather than a mathematical necessity. Crucially, low NRMSE does not imply high Bayesian consistency, as we see for GPT-5 mini.
>   - RMSE/NRMSE are metrics computed using predictions and ground truth stimuli. It does not assume particular forms of behavior.
>   - Bayes-factor evidence compares goodness-of-fit between observer models (Bayesian vs non-Bayesian linear baselines) against LLM responses. A model can have low RMSE yet be better fit by a simple linear rule than by a Bayesian observer. For example a perfect identity-mapper (which maps input stimulus to the right estimate every time), would achieve minimal RMSE but exhibit no regression to the mean or Kalman-like sequential behavior
>
> Furthermore, we believe standalone goodness-of-fit is insufficient as evidence of “being Bayesian,”. While we report Bayes factor as an observation, BayesBench rely primarily on BCS as a behavioral measure. BCS asks if model adapts in Bayes consistent manner (for example, if we increase measurement noise does model rely more on prior information) and can uncover behavior not detectable with goodness-of-fit in static conditions.
>
> The BayesBench score is defined as a multi-component metric, with both a) capability measures via baseline NRMSE (measuring task accuracy) and Bayes-RRE (measuring accuracy vs a Bayes-optimal combiner, which is our normative baseline), and b) decision-strategy/behavior via BCS (consistency of parameter shifts with Bayes expectation). Our final score breaks down between these components to provide clear attribution.
>
> We have revised the manuscript accordingly and reflected the following changes:
>   - Explicitly define capability (baseline RMSE/NRMSE) and decision strategy (behavior changes, BCS) in the methods section.
>   - Added in section 5.1 clarifications to avoid any implication of circularity between task performance and behavior
>   - Clearly state in 3.2 that the key diagnostic for Bayesian consistent behavior is BCS instead of the goodness-of-fit.
>
> **Weakness 2: Limited evidence for benchmark generalizability**
>
> We agree that cross-task performance should be shown more clearly. BayesBench is meant to provide a behavioural profile across tasks and we expect that some models will be more Bayes-consistent in some experiments than in others; we think this difference in performance is informative in showing conditions under which model behavior changes.
>
> We have revised the manuscript accordingly and reflected the following changes:
>   - Add a table with BCS metrics for each experiment in Appendix A9. Performance varies based on task, for example, Mistral scores highly in the marker location task but poorly in the line length ratio and maze distance tasks; on the other hand, Llama-4 Maverick shows better performance across all tasks
>   - Clarify in 3.6 that aggregate score is a holistic summary, and that BayesBench is intended to characterise models across multiple tasks rather than certify universal Bayes-optimality.

---

> ### Author Response · Authors · 2025-11-24
> **Official Comment by Authors (Part II)**
>
> **Weakness 3: Weak tests due to uniform priors**
>
> The use of simple, range-uniform priors are typical in magnitude-estimation psychophysics studies and allows us to compare with decades of experiments in human psychophysics. Range-uniform priors allow us to clearly isolate prior and likelihood and enable our observer models to be interpretable. In our behavioral measurement, BCS, the emphasis is on how models adapt behavior in response to changes in test conditions, and it is crucial that our observer model can have a clear interpretable form.
>
> We also wish to clarify that our range-uniform prior is designed such that for each experiment there are three distinct sessions, and for each session a different uniform prior is used. It is not a global priorr; models must adapt to each particular session, which allows us to test for Bayes-consistent behaviour.
>
> We have revised the manuscript accordingly and reflected the following changes:
>   - Clarify in section 4 the rationale of using session specific range-uniform priors per task, and explain why this is helpful for interpretability.
>   - Add in the Discussion that for future work, richer prior structures (e.g. skewed or bimodal stimulus distributions) can be a natural extension for these studies.
>
> **Weakness 4: Insufficient evaluation of non-Bayesian alternative explanations**
>
> As noted in the segment on weakness 1, because all our tasks are identity mappings, this means that a broad class of simple non-probabilistic heuristics actually collapse to monotone transformations plus noise (e.g., fixed biases, gain distortions, regression to an anchor, log-transform compressions). Our set of unconstrained linear models (including versions with a log-transform before the linear mapping) already capture them.
>
> For our cue combination metric (RRE), our goal is not to fit a Bayesian observer to the data. Instead, we are comparing LLM's multimodal response against a sensible normative baseline. This baseline is a Bayes-optimal linear combination of the same model’s unimodal responses. Bayes-RRE measures how close the LLM's multimodal prediction accuracy is, against this normative baseline. The idea is:
>   - A reasonable baseline is to linearly combine unimodal estimates with reliability-weighted weights assuming Gaussian noise. This is also the model followed in many biological systems
>   - Bayes-RRE checks whether LLMs can at least match that level of accuracy; in practice only Llama-4 Maverick matches / slightly exceed this. This is an interesting discovery because it suggest Llama-4 Maverick can utilize non-linear patterns, as we demonstrate in Fig 6 (a non-linear random forest fits the multimodal vs unimodal responses better than linear variants), which surpasses the model used by biological systems including human. This also shows LLMs that have not yet reached saturation as having room for further improvements.
>
> Our aim in this work is to study if behavior under ablations changes in a Bayes-consistent and when multimodal LLMs combine cues, how they perform relative to a Bayes-optimal linear combiner baseline. Knowledge of both can help guide future model development.
>
> We have revised the manuscript accordingly and reflected the following changes:
>   - Clarify in 3.2 the definition of our alternative baselines and summarise their behaviour relative to Bayesian variants.
>   - Clarify in 3.5 that Bayes-RRE is an accuracy-based comparison to a Bayes-optimal linear combiner, not a behavior fit, and that we interpret it as performance measure against a reasonable normative strategy that has a clear probabilistic interpretation and biological origin.
>   - Expand the Discussion with a short paragraph acknowledging more complex heuristics (e.g. salience-based or switching rules), and making clear that our claims are about Bayes-consistent behavior under a simple observer model, not a proof that no heuristic model can reproduce these patterns.
>
> **Addressing specific questions**
>
> **Q1. Separating capability from decision strategy**
>
> As noted above, the identity-mapping design allows a clean separation between a) capability, via baseline RMSE/NRMSE (and Bayes-RRE). These are measures of how accurately the model reports the presented stimulus and b) decision strategy, via BCS. This summarises how fitted Bayesian parameters change across ablations manipulating prior strength and measurement noise.
>
> In the updated manuscript, we clarified in 3.5 the capability/strategy distinction.
>
> **Q2. Cross-task and cross-modality consistency**
>
> We agree this should be quantified explicitly. In our update we will report segregated BCS for each task. In the updated manuscript we clarified that BayesBench is intended to yield a multi-dimensional profile and a holistic aggregate score. Please see note on weakness 2 in the prior section.

---

> ### Author Response · Authors · 2025-11-24
> **Official Comment by Authors (Part III)**
>
> **Q3. Stronger tests with non-uniform priors**
>
> We agree these would make it easier to dissociate Bayesian observers from certain heuristics when comparing goodness-of-fit. However, our current study focuses on studying changes in LLM behavior when conditions are altered. We believe our range-uniform priors plus controlled ablations (ablations effectively change prior strength and measurement reliability) is a better compromise for simplicity and controllability. Please see further details on weakness 3 in the prior section.
>
> We have revised the manuscript accordingly and reflected the following changes:
>   - Clarify in section 4 our use of range-uniform priors for interpretability and alignment with standard psychophysical designs
>   - Explicitly acknowledge that non-uniform prior tasks are an important extension for future work and that our current claims are restricted to Bayes-consistent integration under simpler range-specific priors and our designed ablations.
>
> **Q4. Evaluation of non-Bayesian computational models**
>
> Please see further details around weakness 4 in the prior section. Given our tasks are identity-mapping, the unconstrained linear baselines already cover a range of standard heuristic relevant to this setting. For cue combination, we do not fit a Bayesian observer; instead we compare model’s multimodal accuracy to a Bayes-optimal linear combiner as a baseline. The cue combination study is more on quantifying model performance relative to this reasonable baseline, rather than on behavioral fitting.
>
> In the updated manuscript, we clarified these design choices and state that a systematic exploration of richer heuristic models is an important avenue for future work, especially once study beyond identity-mapping tasks.
>
> **Q5. Behavioural signatures beyond regression to the mean (sequential models)**
>
> For static tasks, we agree that regression to the mean alone is not uniquely Bayesian, which is why we place emphasis on behavioral change under different ablations (via BCS).
>
> For the sequential task, our observer is a Kalman filter with a non-zero process noise. In such models, posterior variance converges to a steady state rather than decreasing monotonically with trial index. In our experiments we did not see “posterior sharpening” curves of the kind associated with long-term learning of hyperparameters.
>
> We have revised the manuscript accordingly and reflected the following changes:
>   - Clarify in 3.2 that the sequential observer is intended to model within-session inference in a noisy environment, not hierarchical learning of priors over many sessions.
>   - Note in the Discussion that exploring longer-term posterior sharpening and hierarchical learning is an interesting direction for future developments.
>
> **On your further comments:**
>   - Ethics statement: we added an ethics statement (appendix A6) describing the human-subject procedure.
>   - Minimum trials for reliable BCS: We will report the recommended trials per condition for stable estimates.
>   - Beyond regression to mean another hallmark we tested in behavioral fitting is a kalman-filter like behavior. However, as noted above the main emphasis with regards to Bayesian behavior is on changes in parameters as measured by BSC, rather than static goodness-of-fit, which is less robust. In our experiments we found no clear changes in systematic deviation from ground truth with accumulated experience within sessions.
>   - Figure 3 (ground truth): We will add the ground truth curve to Fig. 3.
>   - Handling instruction-following failures: Trials with non-numeric or off-format responses were discarded and will not form part of the context to the next interaction. We have included this in the A 2.2
>   - Figure 4 (identity line): We will add an identity line  to Fig. 4.
>   - “Behavioural modelling” vs “cue-combination modelling”: We will rephrase the relevant passage to state clearly the concepts. Behavioral modelling is in fitting observer models to LLM responses while cue-combination modelling is fitting different cue combination models against observation. However in our study the latter (cue combination modelling) is not as relevant as the RRE, which compares model accuracy against a Bayes linear combiner.
>   - Notation (“LLM” vs “model”): we will standardise and clean up subscripts accordingly
>   - GPT-5 and external tools: tool use was disabled in all experiments; we will state explicitly that models had no access to external calculators or search.
>   - “Bayesian factor evidence” definition: We will define this term clearly in the main text when it first appears. It is as an AIC-based comparison of Bayesian and non-Bayesian models in our set.

---

> > ### Comment · Reviewer_d9eK · 2025-11-26
> >
> > I am happy with the authors' replies to my previous questions and have raised my rating accordingly.

---

### Author Response · Authors · 2025-11-24
**Summary Answers to Common Themes and Main Points Across Reviews**

We thank you all reviewers for their time and thoughtful feedback. We will address each reviewer’s comments in detail below, but first we briefly summarize the main common themes and clarifications here:

- Capability and behavioural strategy are distinct, often dissociated dimensions of model quality
  - We find that high capability (accuracy) does not guarantee Bayesian consistency or Bayesian like uncertainty handling. GPT-5 mini is an example where a highly accurate model does not imply high Bayesian consistency
  - Real-world deployment requires robustness just as much as accuracy, BayesBench thus exposes important gaps where traditional accuracy based benchmarks may overly favour brittle models.

- BayesBench measures both capability and strategy
  - Capability is captured with NRMSE (measures task accuracy) and Bayes-RRE (measures performance relative to Bayesian normative baseline).
  - Behavioural strategy is captured by the Bayesian Consistency Score (BCS), which tracks how models adapt when we manipulate uncertainty.
  - Together, these metrics are more sensitive to Bayesian signatures than simple goodness-of-fit in static tests or accuracy alone

- Bayes RRE uses a principled normative benchmark for cue-combination and Llama-4 Maverick's outperformance is an important observation.
  - The normative baseline used is a Bayesian linear cue combiner assuming Gaussian noise. This is a model widely applied in biological systems including humans.
  - We found Llama-4 Maverick, closely matches and sometimes exceeds this linear baseline; a random-forest fit (figure 6 in our paper) suggests it uses non-linear cue integration, surpassing the known cue integration methods in biological systems including humans
  - Most models fall short of the Bayes-optimal linear combiner, indicating substantial headroom for future development

- Behavioural consistency is task-dependent, there is not yet evidence of universal computation principle
  - Models show different Bayesian consistency scores across tasks: some excel on one task but not others, and none achieve uniformly high consistency across all three. Bayesian-like behaviour is fragmented, and task dependent.
  - We show in Appendix A.9 a BCS breakdown by model and task

- Prompt “steering” ablations reveals model sensitivity to contextual prior information
  - When we inject skewed numeric ranges into the prompt as a prior (we tell models that for a number of past observations, values are seen to fall between X and Y), model errors increase. This shows that models are sensitive to provided prior, and when the provided prior is intentionally skewed and incorrect, they make more error despite its observations.
  - In the panel A figure 6, circles denote base performance and triangles denotes performance when a skewed numerical range is given in the context. The triangles are seen to generally lie further away, indicating larger errors committed.

- Controlled noise manipulations reveal interesting patterns and is a key strength of the BayesBench framework.
  - Injecting modality specific noise allow us to investigate if models down weigh the correct channel
  - In the panel B of figure 6, we plot the change in implied image modality weighing when noise is injected. The dark blue bars show that most models indeed down-weigh images when noise is present. Our framework is designed to isolate and analyze behaviors like these.

- BayesBench framework enables controlled and flexible noise manipulations, which uncovers interesting properties and can be further developed for future use:
  - Biased prompt affects model accuracy
  - Image only noise checks if model re-weigh noisy channel reasonably

---

### Author Response · Authors · 2025-12-03
**Final Author Update**

We thank all reviewers for their thoughtful questions and engagement.

While we have addressed in detail all questions raised by reviewers, we wish to provide an additional high-level summary to assist the new AC's assessment and to provide context on the discussion status prior to the platform reset.

**Status of Reviewer Consensus**

Reviewer d9eK engaged most actively with our rebuttal and explicitly confirmed that our clarifications resolved their concerns. In their final comment (Nov 26), they stated that they had increased their score substantially to reflect the improvements. While the other reviewers did not post final responses before the freeze, we believe our individual responses below and updated manuscript (including new Fig 6 and table 4 in appendix A9) directly address the specific concerns regarding the design choice of our tasks, interpretation of cue-combination performance, prompt sensitivity and potential confounding factors raised by Reviewers 3KTP, sjdc and SQto.

**Summary Response**

In our rebuttals and in our updated manuscript, we have clarified the below main points:

1. BayesBench is a systematic experimental pipeline with controlled ablations that bridge with decade-long human psychophysics studies. Amongst other observations:
- Accuracy and robustness dissociate and the correlation between model-scale, Bayesian behavior and accuracy is nuanced
- Behavioural consistency is task-dependent and there is not yet evidence of a universal computation principle across models and tasks
- Established a flexible framework to apply human psychophysics studies in LLMs and behavioral metrics that will inform future LLM development

2. BayesBench decouples capability and decision strategy
-  NRMSE / Bayes-RRE are capability measures; BCS is a behavioral measure
-  Separation of capability and behavior allows a detailed breakdown of performance and, more importantly, shows that models such as GPT-5 Mini can be highly capable and yet not behave in a Bayes-optimal manner.
-  This highlights that benchmarks that primarily focus on capability (e.g. accuracy) may favour models that are less robust

3. Behavioral evidence is not based on goodness-of-fit alone, it is measured using the Bayes Consistency Score (BCS).
- BCS measures changes in behavior when conditions change.
- This is why we use synthetic tasks under systematic ablation; they allow us to control conditions and detect subtle behavioral changes

4. Llama-4 Maverick finding
-  In the Bayes RRE benchmark, Llama-4 Maverick is found to outperform the normative baseline (a linear Bayesian cue combination model).
-  Reviewers asked if this is problematic; we clarified that Llama-4 outperforming this is significant as it implies the utilization of non-linear cue integration (demonstrated in Fig 6) beyond known biological systems.

We hope this summary assists in your assessment and prediction of the reviewer consensus.

---

### Meta-Review · Area_Chair_vhTr · 2026-01-06

**Summary:**

Reviewers had major concerns about the validity and strength of claims made by the authors. In particular, reviewers were concerned that results could be explained by simpler alternate hypotheses, that some of the results actually provided evidence against the key claim of emergent Bayesian behavior, and that results on the proposed benchmark would not generalize to more complex or naturalistic settings.

**Reviewer Concerns:**

While the authors have agreed to soften some of their claims, the core narrative still focuses on "emergent Bayesian behavior" which reviewers felt was not strongly supported by the presented evidence. The authors have not fully resolved concerns that alternate hypotheses may explain the same results. Furthermore, the authors made additional strong claims in their rebuttal that I believe the reviewers would strongly disagree with ("Llama-4 outperforming this is significant as it implies the utilization of non-linear cue integration (demonstrated in Fig 6) beyond known biological systems").

**Reviewer Scores:**

It seems like reviewer d9ek intended to raise their score, which I would guess they would have raised to a 6 (originally a 4). I do not believe the other reviewers would have changed their scores.

---

### Decision · Program_Chairs · 2026-01-26

Reject